# BALROG: Benchmarking Agentic LLM and VLM Reasoning On Games

**Davide Paglieri**[1][*]**Bartłomiej Cupiał**[2,6][*]**, Samuel Coward**[3]**, Ulyana Piterbarg**[4]**,**
**Maciej Wolczyk**[2]**, Akbir Khan**[1,5]**, Eduardo Pignatelli**[1]**, Łukasz Kuciński**[2,6,7]**, Lerrel Pinto**[4]
**Rob Fergus**[4]**, Jakob Nicolaus Foerster**[3]**, Jack Parker-Holder**[1]**, Tim Rocktäschel**[1]
[1]AI Centre, University College London, [2]IDEAS NCBR, [3]University of Oxford,
[4]New York University, [5]Anthropic, [6]University of Warsaw,
[7]Institute of Mathematics, Polish Academy of Sciences
d.paglieri@cs.ucl.ac.uk

## Abstract

Large Language Models (LLMs) and Vision Language Models (VLMs) possess extensive knowledge and exhibit promising reasoning abilities, however, they still struggle to perform well in complex, dynamic environments. Real-world tasks require handling intricate interactions, advanced spatial reasoning, long-term planning, and continuous exploration of new strategies—areas in which we lack effective methodologies for comprehensively evaluating these capabilities. To address this gap, we introduce BALROG, a novel benchmark designed to assess the agentic capabilities of LLMs and VLMs through a diverse set of challenging games. Our benchmark incorporates a range of existing reinforcement learning environments with varying levels of difficulty, including tasks that are solvable by non-expert humans in seconds to extremely challenging ones that may take years to master (e.g., the NetHack Learning Environment). We devise fine-grained metrics to measure performance and conduct an extensive evaluation of several popular open-source and closed-source LLMs and VLMs. Our findings indicate that while current models achieve partial success in the easier games, they struggle significantly with more challenging tasks. Notably, we observe severe deficiencies in vision-based decision-making, as several models perform worse when visual representations of the environments are provided. We release BALROG as an open and user-friendly benchmark to facilitate future research and development in the agentic community. Code and Leaderboard at balrogai.com.

## 1 Introduction

Recent successes of Large Language Models (LLMs) have renewed interest in building general-purpose agents capable of autonomously achieving complex goals Yang et al. (2023). LLMs possess vast knowledge across domains (Brown, 2020; Hendrycks et al., 2020), can reason in specific scenarios (Wei et al., 2022a; Shinn et al., 2023; Rein et al., 2023), and can reliably follow human instructions in simple settings (Ouyang et al., 2022). These abilities suggest that LLMs have the potential to become efficient *agents*, capable of autonomously performing a wide range of human tasks that require sequential decision making. In the present day, however, state-of-the-art models continue to exhibit persistent failure modes on many of the skills that are crucial for autonomous real-world interaction. For example, LLMs fail to act robustly in dynamic environments, and they cannot reliably learn from mistakes, reason about space and time, or plan over long time horizons (Xing et al., 2024; Yamada et al., 2023; Kambhampati et al., 2024). Improving our understanding of LLM capabilities through rigorous, safe evaluations is key for assessing the risks and limitations of deploying agentic LLMs in the real world.

Current agentic benchmarks evaluate LLM performance in settings that involve no more than a few dozen rounds of interaction between a model and an environment, e.g., solving simple office

---

[*]Equal contribution.

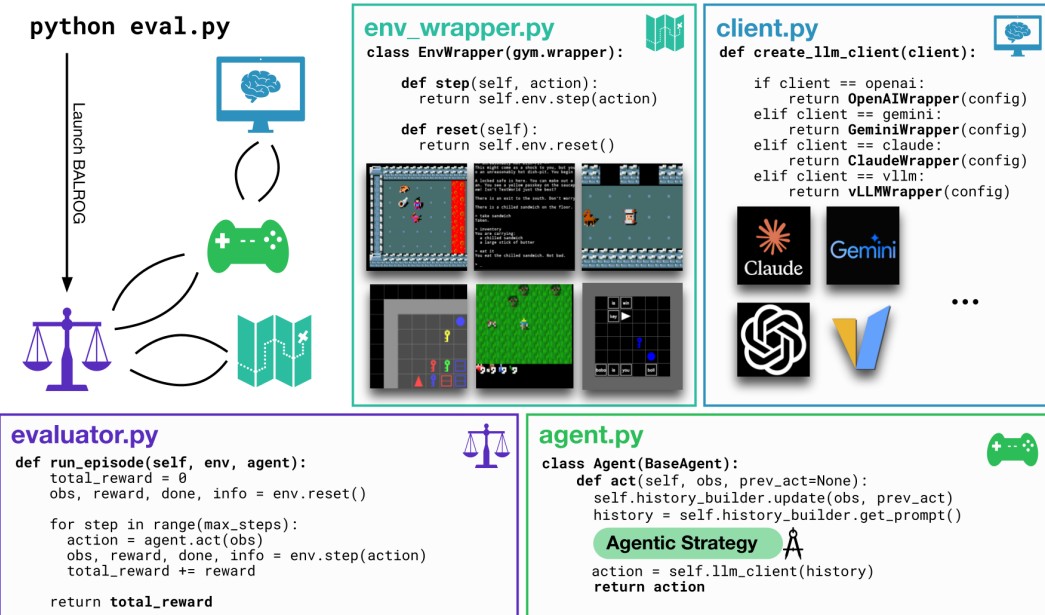

Figure 1: **An overview of the BALROG Benchmark for evaluating LLMs on long-context interactive tasks**. Submissions of new inference-time methods for improving the capabilities of an existing model via an "agentic strategy" need only modify the `agent.py` file. Similarly, benchmarking a new model zero-shot can be done by adjusting a configuration file in `client.py`. The agent class includes a prompt builder to manage observation history, and a client that abstracts the complexities of various APIs and model-serving frameworks. The `env_wrapper.py` file standardizes interaction across settings, and the evaluator executes agents and collects performance metrics.

tasks (Wang et al., 2024), navigating the Internet (Zhou et al., 2023), and resolving GitHub issues (Jimenez et al., 2023). New agentic prompting frameworks and improvements to short-horizon reasoning via LLMs like OpenAI o1 have led to dramatic and fast-paced gains in state-of-the-art performance on these benchmarks (OpenAI, 2024b; Wang et al., 2023; Fernando et al., 2023; Hu et al., 2024). However, many realistic tasks require orders of magnitude more interactions (Pignatiello et al., 2020; Wansink & Sobal, 2007).

In this paper, we argue that the next frontier for language and vision-language model capabilities lies in long-horizon reasoning and decision-making. To that end, we propose BALROG: Benchmarking Agentic LLM and VLM Reasoning On Games. BALROG is a benchmark and framework that aggregates a diverse set of complex reinforcement learning game environments into a unified testbed for research on long-context LLMs. Games have historically served as highly effective metrics for evaluating progress in deep reinforcement learning research (Bellemare et al., 2013; Silver et al., 2018; Schrittwieser et al., 2020; Vinyals et al., 2019). By aggregating many different game environments into a single evaluation, we look to spur progress on developing truly generalist agents that can meaningfully address embodied, real world tasks. Specifically BALROG enables seamless running of LLM and VLM agents on BabyAI, Crafter, TextWorld, Baba Is AI, MiniHack, and NetHack (Chevalier-Boisvert et al., 2019; Hafner, 2021; Côté et al., 2019; Cloos et al., 2024; Samvelyan et al., 2021; Küttler et al., 2020). These environments have lightweight simulators, ensuring that the benchmark is affordable for the research community. Furthermore, while all of these games are long-horizon, they span a broad range of difficulty levels, from tasks where we see fair zero-shot performance by state-of-the-art long-context models (BabyAI) to those where even specialized neural models trained on billions of in-domain datapoints make very limited progress (NetHack) (Piterbarg et al., 2024; Klissarov et al., 2023; Wołczyk et al., 2024). BALROG is difficult to solve through simple memorization – all of the environments used in the benchmark are procedurally generated, and encountering the same instance of an environment twice is unlikely.

Using the six proposed environments, we evaluate the capabilities of various popular LLMs and VLMs. We employ a fine-grained metric that captures how close each model is to completing a task, which gives us a thorough understanding of the resulting trajectories. In our qualitative analysis, we

study the agents' capabilities for spatial reasoning, systematic exploration, long-term planning, and discovering environment dynamics. We find that the current top LLMs show promise on the simplest tasks but completely fail to make meaningful progress on the more difficult tasks, such as MiniHack and NetHack. Some of the models exhibit knowledge about the game from pre-training but fail to use it in practice. For example, in NetHack, GPT-4o often dies from the consumption of rotten food, even though, when prompted, it correctly identifies it as very dangerous. Furthermore, we study the impact of the input representation. Although the majority of the environments were created with vision in mind, we find that multimodal LLMs perform much worse when also presented with an image of the environment rather than a textual-only description of the observation. This suggests that reliable vision-based decision-making is currently far outside our reach.

Our results show that BALROG is a very difficult benchmark that still allows us to observe fine-grained progress in crucial areas such as long-term planning, spatial reasoning and navigation. We share the codebase and open the benchmark for external submissions. We summarize our contributions as follows:

- BALROG, a suite of six reinforcement learning environments for testing the agentic capabilities of long-context LLMs. We provide a fine-grained metric for model evaluation, and we develop a novel data-informed progression system for NetHack.
- Baseline evaluations of state-of-the-art LLMs on BALROG using zero-shot prompting, in both Language-Vision and Language-only modalities. We show that while models exhibit decent performance on easier games, all are very far from solving the hardest game in the benchmark, NetHack. We observe that the performance drops further when images of the environment are presented, suggesting severe problems with VLM decision-making.
- We perform a qualitative analysis of the results across capabilities such as spatial reasoning, systematic exploration, and long-term planning. We identify an intriguing knowing-doing gap where the models cannot employ the knowledge they possess.
- An open-source toolkit for benchmarking long-context models on BALROG. This toolkit enables researchers and practitioners to quickly evaluate model performance. While the baseline evaluations performed in this paper are zero-shot, the BALROG toolkit supports inference-time prompting strategies like chain-of-thought (Wei et al., 2022b), few-shot learning, and more.

## 2 BALROG

BALROG is a benchmark and framework that aims to improve our understanding of whether existing long-context LLMs are agentic, i.e., whether they can be used to automate complex activities that require sequential decision-making. It supports model evaluation on challenging reinforcement learning environments that test skills such as long-term planning, spatial reasoning, and the ability to deduce the mechanics of the environment.

By design, the BALROG framework explicitly decouples inference-time prompting strategies from underlying models. The goal of this design choice is two-fold: (1) to facilitate rapid prototyping of inference-time methods for improving model performance on long-context decision-making beyond zero-shot prompting and (2) to ensure that model evaluations are consistent and rigorous.

In the remainder of this section, we introduce the game environments evaluated in the benchmark and we discuss our protocols for model submission to the BALROG Benchmark Leaderboard[1].

### 2.1 ENVIRONMENTS

BALROG evaluates long-context models as agents on the games described below.

**BabyAI.** (Chevalier-Boisvert et al., 2019; Carta et al., 2023) A simple, two-dimensional grid-world in which the agent has to solve tasks of varying complexity described in natural language (e.g., "go to the blue ball, then pick up the grey key"). Agents are tested across five different types of navigation tasks, see Appendix A.

---

[1]This Leaderboard will open to the public at the time of publication.

Table 1: **The tested skills, time horizons, and complexities of interactive decision-making tasks evaluated in BALROG**. Compared to existing benchmarks, BALROG provides infrastructure for evaluating model reasoning and decision-making on harder, longer time-horizon interactive settings. The evaluated tasks span a range of difficulties.

| Skills | BabyAI | TextWorld | Crafter | Baba Is AI | MiniHack | NLE |
|---|---|---|---|---|---|---|
| Navigation | ✔ | ✔ | ✔ | ✔ | ✔ | ✔ |
| Exploration | ✔ | ✔ | ✔ | ✔ | ✔ | ✔ |
| Resource Management | ✘ | ✔ | ✔ | ✘ | ✔ | ✔ |
| Complex Credit Assignment | ✘ | ✘ | ✔ | ✔ | ✔ | ✔ |
| Deducing Env. Dynamics | ✘ | ✘ | ✘ | ✔ | ✔ | ✔ |
| Long-term Planning | ✘ | ✘ | ✘ | ✔ | ✔ | ✔ |
| Turns to Complete | $10^1$ | $10^2$ | $10^3$ | $10^2$ | $10^2$ | $10^4$–$10^5$ |
| Time to Master for Humans | Seconds | Minutes | Hours | Hours | Hours | Years |

**Crafter.** (Hafner, 2021) A Minecraft-inspired grid environment where the player has to explore, gather resources and craft items to ensure their survival. Agents are evaluated based on the number of achieved milestones, such as discovering new resources and crafting tools, see Appendix B.

**TextWorld.** (Côté et al., 2019) An entirely text-based game with no visual component, where the agent has to explore mazes and interact with everyday objects through natural language (e.g., "cook potato with oven"). Unlike the other environments in BALROG, TextWorld is not a grid-world. Models are evaluated on three different tasks, see Appendix C.

**Baba Is AI.** (Cloos et al., 2024) An environment based on the popular puzzle video game *Baba Is You*. The player manipulates the rules of the game world by pushing word blocks, altering how objects interact. Agents are tested on 40 puzzles, see Appendix D.

**MiniHack.** (Samvelyan et al., 2021) MiniHack is a multi-task framework built on top of the NetHack Learning Environment (Küttler et al., 2020). We select five different tasks, Maze, Corridor, Corridor-Battle, Boxoban, and Quest. Collectively, they assess a wide range of skills, including exploration, navigation, long-term planning, and resource management, see Appendix E.

**NetHack Learning Environment (NLE)** (Küttler et al., 2020) is based on the classic roguelike game NetHack, known for its extreme difficulty and complexity. Success in NetHack demands both long-term strategic planning, since a winning game can involve hundreds of thousands of steps, as well as short-term tactics to fight hordes of monsters. Accurate credit assignment is also crucial to understanding which actions contributed to success or failure. It takes human players years to master NetHack without accessing external guides. Notably, we find that research shows that LLMs can answer questions about the game mechanics and optimal strategies (see Appendix F.5), but they fail to apply this knowledge in practice. See Appendix F for more details.

Table 1 provides an overview of the environments used in the benchmark, detailing the reasoning and agentic capabilities required to succeed in each. This diverse set of environments positions BALROG as a comprehensive benchmark for assessing the capabilities of LLM agents, making it a valuable tool for evaluating their performance for years to come.

## 2.2 SUBMITTING TO THE BENCHMARK LEADERBOARD

The BALROG benchmark accepts two types of submissions.

**New Models.** Submissions may include any type of new model, such as large language models (LLMs), vision-language models (VLMs), large-action models (LAMs), or fine-tuned versions of existing models. The key requirement is that these models must be capable of generating actions in natural language. By default, these models will be evaluated zero-shot.

**Agentic Strategies.** Submissions may propose novel inference-time prompting strategies for improving the reasoning, planning, or in-context learning capability of an existing model. These strategies should extend beyond simple zero-shot prompting for direct action prediction, demonstrating more sophisticated techniques for inference-time decision-making.

## 3 ZERO-SHOT EVALUATION PROTOCOL

In this section, we provide a description of our protocols for evaluating state-of-the-art, long-context LLMs and VLMs on BALROG. These evaluations are intended to serve as baselines for the benchmark. As a result, they probe zero-shot performance only.

### 3.1 EVALUATION SETTING

We aim to keep the evaluation setting simple. During each timestep of interaction, agents are prompted to output the next action as a natural language string, conditioned on their past interaction history in the environment. To perform successfully in BALROG, models must demonstrate robust instruction-following capabilities, including reading and interpreting game rules, understanding the action space, and producing valid actions to complete tasks effectively.

To address cases where the LLMs/VLMs output hallucinated or invalid actions, AgentQuest provides feedback to the agent indicating the action's invalidity, it then executes a default fallback action (such as a "do-nothing" action or a standard move like "north"), and logs the occurrence for trajectory statistics. This ensures that the interaction remains continuous and robust while enabling users to analyze the context and frequency of such errors in post-evaluation analysis.

A diagrammatic visualization of BALROG is shown in Figure 1. We conceptualize the agent as a combination of the underlying LLM/VLM model and a particular prompting strategy. We provide a unified client wrapper that seamlessly integrates APIs for closed-source LLMs and VLMs such as OpenAI, Gemini, and Claude and allows users to effortlessly switch and evaluate models. For the evaluation of locally-served models, we include native support for the vLLM library (Kwon et al., 2023), which optimizes throughput by efficiently batching generation requests. We use multiple seeds for each environment to ensure the statistical significance of the results.

**Metrics** To ensure a fair and interpretable evaluation, we introduce a standardized metric, scoring performance on each task within a range of 0 to 100. For environments like MiniHack, BabyAI, and Baba Is AI, each episode is scored as either 0 or 100 based on task completion. For TextWorld, Crafter, and NetHack we use as the score a real number between 0 and 100, representing the proportion of achievements toward the maximum score. For NetHack, as the game scoring system does not adequately reflect actual progression (Wołczyk et al., 2024), we propose a novel, data-informed progression metric, described in Appendix F.2, to better capture agent performance.

**Performance** BALROG supports highly parallelized evaluations, leveraging the lightweight simulators of each of the environments in the suite. These evaluations allow multiple agents and environment instances to run concurrently with minimal computational overhead. Environment instances run asynchronously from one another, accommodating varying observation lengths and ensuring that agents with faster generation speeds (per action) are not affected by slower agent bottlenecks.

### 3.2 OBSERVATIONS

In the initial prompt, the agent is introduced to the game rules and provided with a list of available actions, each accompanied by a brief description. To prevent model overspecialization, we design a general prompt that is not fine-tuned to any specific LLM. Subsequent prompts present the observation-action history in a chat-based format. The game rules and observations are conveyed from the perspective of the "user", while prior actions are attributed to the "assistant" or "model" role, depending on the type of model used. This structure mirrors the standard format used for fine-tuning instruction-following LLMs. Detailed examples of game observations are included in the appendices.

Except for TextWorld, which lacks a visual component, we evaluate all environments using two observation modalities:

**Language Only Format** Observations are expressed as natural language descriptions of the environment's state (e.g., "a wall 5 steps ahead, a wall 2 steps to the left. . . "). For environments without native textual representations, we either generate descriptions using open-source language wrappers (BabyAI (Carta et al., 2023), Crafter (Wu et al., 2023), NetHack, and MiniHack (Goodger et al., 2023)) or develop a custom wrapper ourselves (Baba is AI, see Appendix D)

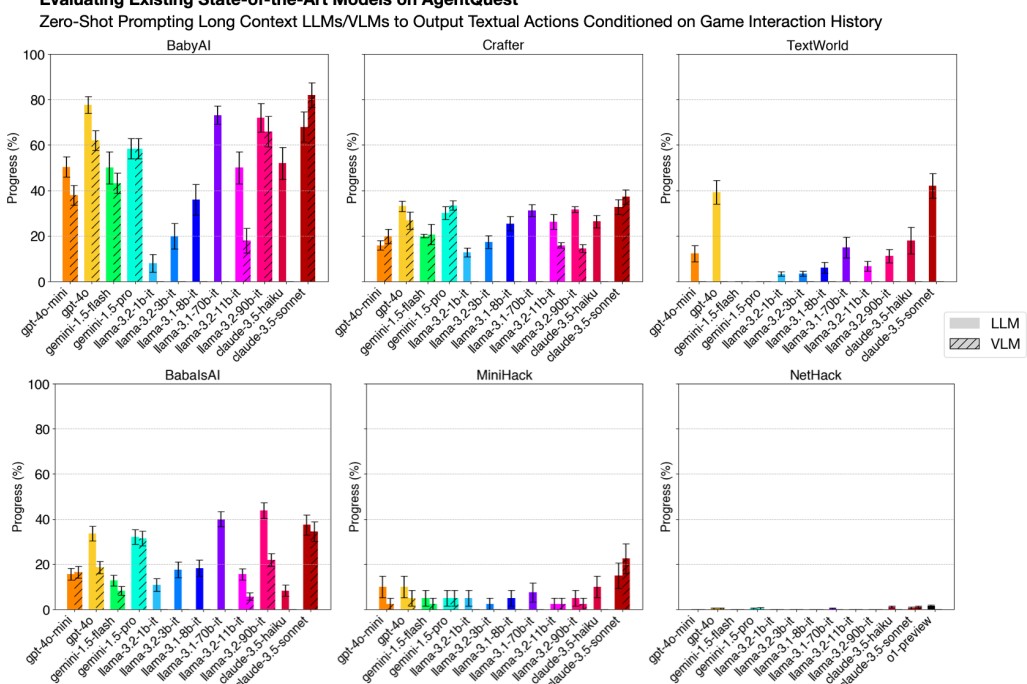

Figure 2: **Baselines for BALROG**. We evaluate the zero-shot performance of seven state-of-the-art and long-context LLMs and VLMs on BALROG. During each timestep of interaction, models are prompted to output the next in-game action conditioned on past interaction history. Standard error is obtained by running multiple replicate seeds, as detailed in the Appendix.

**Vision-Language Format** For VLMs, the observation consists of an image representing the environment's current state, alongside its natural language description (mentioned above). In this format, the image corresponds only to the current observation, although we support including multiple images in the observation history.

For the most complex environments, i.e., MiniHack and NetHack, we augment the language-based observations with a two-dimensional map rendered using ASCII characters. For all experiments, we use a history length of 16 observations to maintain consistency across tasks. However, participants submitting to this benchmark are allowed to modify the observation history length as needed for their respective models and experiments.

## 3.3 MODELS

We evaluate a range of popular closed-source and open-source models, including Gemini-1.5-Flash and Gemini-1.5-Pro (Reid et al., 2024), GPT-4o-mini (2024-07-18 release) and GPT-4o (2024-05-13 release) (Achiam et al., 2023; OpenAI, 2024a), Claude 3.5 Sonnet and Claude 3.5 Haiku (2024-10-22 releases) (Anthropic, 2024), as well as Llama 3.1 instruct (8B and 70B) (Dubey et al., 2024) and Llama 3.2 instruct (1B, 3B, 11B and 90B) (MetaAI, 2024). Additionally, we test o1-preview (2024-09-12 release) (OpenAI, 2024b) exclusively on the NetHack environment due to budget constraints.

## 4 RESULTS

In Figure 2, we present the results of our experiments using the BALROG evaluation script for both language-only and vision-language formats. Most leading models demonstrate fair average progression on BabyAI, Crafter, and Baba Is AI, with GPT-4o and Claude 3.5 Sonnet performing best. Interestingly, the open-source Llama 3.1 70B and Llama 3.2 90B models achieve the highest results on the Baba Is AI language-only format, narrowly surpassing GPT-4o and Claude 3.5 Sonnet. In TextWorld, GPT-4o and Claude 3.5 Sonnet lead, while Gemini models fail to complete any tasks,

being flagged as 'unsafe' by the Google Gemini API, despite the prompts containing no actual safety concerns. The MiniHack suite proves very challenging for all models, especially the quest and boxoban tasks, which were never solved by any model. Finally, all models flat line with NetHack, with the best-performing model, o1-preview, achieving a meager 1.5% average game progression.

Table 2 summarizes the aggregated results across all environments in the language-only format. Overall, Claude 3.5 Sonnet is the best-performing model, with an average progression of 32.64%, followed closely by GPT-4o and Llama 3.1 70B a few points behind. Gemini-1.5-Pro lags behind the other large models, partly due to its 0% performance on TextWorld. However, results differ for the vision-language format, as shown in Table 3. Here, we observe that both GPT-4o and Llama 3.2 exhibit a decline in performance when image observations are included, likely due to confusion arising from the added visual input. In contrast, Gemini-1.5-Pro and Claude 3.5 Sonnet especially, maintain consistent performance across both formats. This suggests that current multimodal Transformer architectures are still better equipped at handling textual information than visual input, a topic we explore further in Section 6. We show more detailed results for each environment in their appendices.

<table>
<tr><td colspan="2">Table 2: Language-Only Performance</td><td colspan="2">Table 3: Vision-Language Performance</td></tr>
<tr><td>Model</td><td>Average Progress (%)</td><td>Model</td><td>Average Progress (%)</td></tr>
<tr><td>claude-3.5-sonnet</td><td>$32.64 \pm 1.93$</td><td>claude-3.5-sonnet</td><td>$35.48 \pm 2.02$</td></tr>
<tr><td>gpt-4o</td><td>$32.34 \pm 1.49$</td><td>gemini-1.5-pro</td><td>$25.76 \pm 1.36$</td></tr>
<tr><td>llama-3.1-70b-it</td><td>$27.88 \pm 1.43$</td><td>gpt-4o</td><td>$22.56 \pm 1.44$</td></tr>
<tr><td>llama-3.2-90B-it</td><td>$27.29 \pm 1.44$</td><td>llama-3.2-90B-it</td><td>$20.99 \pm 1.58$</td></tr>
<tr><td>gemini-1.5-pro</td><td>$21.00 \pm 1.18$</td><td>gpt-4o-mini</td><td>$15.36 \pm 1.29$</td></tr>
<tr><td>claude-3.5-haiku</td><td>$19.32 \pm 1.83$</td><td>gemini-1.5-flash</td><td>$14.94 \pm 1.40$</td></tr>
<tr><td>gpt-4o-mini</td><td>$17.36 \pm 1.35$</td><td>llama-3.2-11B-it</td><td>$8.43 \pm 1.26$</td></tr>
<tr><td>llama-3.2-11B-it</td><td>$16.82 \pm 1.47$</td><td></td><td></td></tr>
<tr><td>llama-3.1-8b-it</td><td>$15.14 \pm 1.55$</td><td></td><td></td></tr>
<tr><td>gemini-1.5-flash</td><td>$14.63 \pm 1.37$</td><td></td><td></td></tr>
<tr><td>llama-3.2-3B-it</td><td>$10.13 \pm 1.28$</td><td></td><td></td></tr>
<tr><td>llama-3.2-1B-it</td><td>$6.65 \pm 1.04$</td><td></td><td></td></tr>
</table>

## 4.1 QUALITATIVE ANALYSIS

We conducted an analysis of the model trajectories across the environments to identify common behaviors and challenges specific to each setting.

**Spatial Reasoning** While language models demonstrate some proficiency in basic navigation, they exhibit significant limitations in more complex spatial reasoning tasks. In the BabyAI suite, we observed significant shortcomings in the agents' ability to place objects adjacent to other objects, which is required in some scenarios. In NetHack and MiniHack CorridorBattle, good spatial reasoning is crucial during combat, as players need to maneuver within confined corridors to avoid being surrounded by monsters. However, the agents frequently ended up cornered.

**Systematic Exploration** Our experiments revealed a significant weakness in the models' ability to explore. In TextWorld's Coin Collector, where agents must explore a house to locate a coin, agents often wander aimlessly, revisiting rooms they've already explored while missing important areas entirely. An efficient agent would behave in DFS-like manner, methodically searching each room, keeping track of visited areas and prioritizing unexplored spaces. The more complex quests in MiniHack expose similar issues, with models failing to efficiently navigate maze-like structures.

**Long-term planning** The agents exhibit substantial deficiencies in devising and executing long-term plans. We observe near-zero performance on MiniHack, and NLE, which both require careful planning. In particular, we do not observe a single successful trajectory in the Boxoban logical puzzles in MiniHack, which requires careful planning at every step in order to avoid irreversible failures. LLMs, with the finite amount of compute available to them in a single forward pass, are necessarily confined to solving some subset of reasoning problems. We observe that with the current models' depth, number of flops, and reasoning solution templates embedded in the weights, these models cannot solve the reasoning tasks in BALROG. We see a notable improvement with OpenAI

o1's chain of thought capabilities on NetHack, performing close to three times better than its closest competitor in language-only mode Claude-3.5-Sonnet. However, its average progression of 1.57% is still far from satisfactory.

**Discovering and Leveraging Environment Dynamics** Some games require inferring non-trivial causal structure through experimentation to come up with new strategies. For example, a player might identify a `potion of paralysis` by drinking it, and then realize they can use this strategically by throwing such potions at enemies to incapacitate them. This kind of experimentation and strategic thinking is crucial for success in NetHack. However, current models struggle to formulate and execute such context-dependent strategies. In MiniHack Quests environments, models fail to devise and implement multi-step strategies, such as utilizing `wand of cold` or `ring of levitation` to cross lava rivers. In Crafter, where agents can handle basic tasks such as collecting wood, crafting items, drinking water, and even engaging in combat, they fail to learn long-term survival skills such as building shelters for protection against nocturnal threats.

**Knowing-Doing Gap** We observe a pronounced "knowing-doing" gap, where models execute undesirable actions during gameplay despite knowledge of their negative consequences. For instance, in NetHack, models often exit the dungeon shortly after starting the game, resulting in an instant game termination. When queried in a separate thread about the consequences of exiting the first level in NetHack, they correctly identify that it results in an instant death, making it is a highly undesirable action. Similarly, although the models correctly identify that eating rotten food in NetHack can result in death, this remains a common cause of failure, underscoring a disconnect between knowledge and decision-making. Additionally, models tend to ignore even the hints directly present in the input prompt and die from overeating even when advised against it. To study this problem in more detail, we prepared a questionnaire probing basic NetHack knowledge (see Appendix F.5).

## 5 RELATED WORK

The evaluation of large language models has historically relied on benchmarks that emphasize static, non-interactive tasks. Benchmarks such as SuperGLUE (Wang et al., 2019), which tests general-purpose language understanding and MMLU (Hendrycks et al., 2020), which measures massive multitask language understanding, have been instrumental in advancing LLM research. BigBench (Srivastava et al., 2022) further expands the scope by including a diverse set of linguistic and cognitive challenges. Mathematical reasoning datasets like GSM8K and MATH (Cobbe et al., 2021; Hendrycks et al., 2021) assess models' abilities to solve grade-school and competition-level math problems, while Shi et al. (2022) explore multilingual chain-of-thought reasoning. In the domain of code understanding and generation, benchmarks such as HumanEval (Chen et al., 2021) and CodeXGLUE (Lu et al., 2021) evaluate models capabilities in programming tasks.

These benchmarks, however, are limited to single-turn or short-context scenarios, do not require sequential decision-making or adaptation to changing environments and have been saturating rapidly (Kiela et al., 2021). Static benchmarks may not fully capture the progress we are seeking, since the research community aims to push the frontier of agentic foundation models capable of acting in dynamic environments, using tools, planning ahead, and reasoning about their surroundings. Researchers have recently investigated how LLMs use these skills to solve practical tasks, including using computer interfaces to perform office-related chores (Wang et al., 2024; Qin et al., 2024), navigating web pages (Yao et al., 2022; Zhou et al., 2023), and solve GitHub issues (Jimenez et al., 2023). Several works studied the multi-agent capabilities of LLMs to see if they can co-operate (Gong et al., 2023; Piatti et al., 2024) or effectively play against other agents (Jin et al., 2024; Wu et al., 2024).

In this work, we study agentic skills in the context of video games, as they offer challenges well-tailored for human players and test skills that are useful for embodied agents. Previously, some related works employed games to benchmark LLMs (Liu et al., 2023b; Todd et al., 2024; Wu et al., 2023; Ruoss et al., 2024), highlighting their emphasis on problem-solving, spatial reasoning, and well-defined rules and objectives. Some of these benchmarks, however, are already reaching saturation, with environments like Crafter being the most challenging in their suite. In contrast, BAL-ROG fills an important gap by providing a wide range of games at varying difficulties—including the NetHack Learning Environment (Küttler et al., 2020), which takes humans years to master, and where zero-shot LLMs struggle greatly, as also seen in prior work (Jeurissen et al., 2024). These tasks represent a rich and granular testbed for evaluating agentic foundation models, push-

ing decision-making evaluations of LLMs/VLMs to the very limit of their context lengths. Other environments such as MineDojo (Fan et al., 2022) and MineRL (Guss et al., 2019) also present open-ended challenges for agentic capabilities, their steep computational requirements and reliance on multimodal inputs make them less practical for accessible, large-scale benchmarks.

While BALROG currently focuses on evaluating single-agent foundational capabilities, future extensions could explore multi-agent collaboration environments that provide unique opportunities to test teamwork and coordination skills in LLMs. For example, Overcooked (Carroll et al., 2019; Liu et al., 2023a) simulates a cooperative cooking environment where agents must collaborate efficiently under time constraints and task dependencies, testing planning and communication abilities. Another compelling environment is Hanabi (Bard et al., 2020), a cooperative card game where players must rely on indirect communication and inferential reasoning to achieve a shared objective under partial observability. These environments present rich opportunities to benchmark advanced collaboration and multi-agent decision-making skills, which are essential for broader deployment of agentic LLMs.

## 6 OPEN RESEARCH PROBLEMS

Aside from its utility for model evaluations, BALROG also offers a test-bed for rapidly prototyping new inference-time methods for improving the agentic capabilities of LLMs and VLMs. There are many open research problems in this space. As of the writing of this paper, some of the most performant methods for improving model reasoning capabilities on short-form and/or shorter-context problems are infeasible to apply naively to BALROG due to the extremely long-context nature of tasks. Addressing these challenges could further enhance the development of stronger autonomous agents. We highlight several key areas for future work below.

**In-Context Learning and Few-Shot Prompting**  BALROG enables evaluation of In-Context Learning (ICL) agents, which can use few-shot examples to adapt to out-of-distribution tasks. We provide a small dataset of human demonstrations for each environment and an implementation of few-shot conditioning in the BALROG codebase. The benchmark codebase also supports the study of In-Context Reinforcement Learning (Lee et al., 2024; Laskin et al., 2022; Lin et al., 2023), where agents learn to improve from mistakes during inference. On the large models benchmarked in Section 4, naive few-shot learning (i.e., prompting LLM and VLM agents with examples of full human games in-context) is extremely computationally expensive to run on BALROG. For example, a single demonstration of NetHack game-play can require upwards of $700,000$ input tokens to represent in a prompt. Despite advancements in fast inference technologies like caching and falling API costs for long-context prompting, we found these experiments to be infeasible to conduct at this time. Sub-selecting only the relevant parts of demonstrations via retrieval-augmented few-shot prompting strategies (Lewis et al., 2020) might offer a way to circumvent these challenges. We leave exploration of such methods for future work.

**Advanced Reasoning Strategies**  Beyond simply prompting LLMs and VLMs to directly predict the next action of game-play, BALROG also supports the study of more advanced reasoning techniques like chain-of-thought (Wei et al., 2022b), self-refinement (Madaan et al., 2024), and basic planning. These methods have been demonstrated to improve model performance on shorter-context problems. We believe them to be an exciting direction for future work on long-context reasoning and decision-making. For example, model performance on the tasks in BALROG might be improved by integrating multi-agent collaboration (Chang, 2023; Khan et al., 2024; Yao et al., 2024) and tool usage (Shen et al., 2024; Ruan et al., 2023; Schick et al., 2024; Qin et al., 2023) in decision-making. Additionally, incorporating memory mechanisms or reinforcement learning techniques could help bridge the "knowing-doing" gap, enabling models to apply their knowledge effectively in practical, long-horizon tasks. Finally, experimenting with open-ended self-improvement loops (Wang et al., 2023; Hu et al., 2024) could lead to more adaptive and general agents (Team et al., 2023; Hughes et al., 2024), offering a pathway toward truly autonomous systems.

**Limitations of Current Vision-Language Models**  Despite their potential, our benchmark shows significant variability in VLM performance. While some models, like Llama 3.2, struggle to integrate visual information into coherent decision-making, others—most notably Sonnet 3.5—demon-

strate stronger performance in VLM mode. This disparity highlights significant variability in VLM capabilities, which may stem from differences in training objectives and datasets. For example, Sonnet 3.5's superior performance can be attributed in part to its training on tasks involving computer usage (Anthropic, 2024), which inherently require integrating visual and textual inputs for action-based reasoning.

Recent studies have identified key limitations of VLMs that align with our findings, including biases toward natural image-text pairs, optimization for image description rather than action-oriented reasoning, and challenges with out-of-distribution inputs (Tan et al., 2024; Tong et al., 2024; Rahmanzadehgervi et al., 2024; Zang et al., 2024; Guan et al., 2023). These limitations are further exemplified in our benchmark, where grid-based image observations differ significantly from the natural image-text pairs on which many VLMs are trained (Yu et al., 2023; Rahmanzadehgervi et al., 2024). Moreover, the computational cost of image processing constrained our evaluation to a single image per observation, with the remainder of the history provided in text. While this constraint may hinder performance for some models, our results show that certain VLMs like Claude 3.5 Sonnet can still perform robustly under these conditions.

To address these challenges, our codebase already supports multi-image observation histories, and future iterations will incorporate video observations, which are likely better suited for the long-horizon sequential decision-making tasks central to our benchmark. These enhancements aim to better evaluate and leverage the potential of VLMs in complex reasoning scenarios. We plan to introduce support for video observations once prominent models with efficient video-processing capabilities become available, ensuring that our benchmark remains aligned with the latest advancements in VLM technology.

**Computational Limitations of Large Language Models**    Mechanistic interpretability could provide valuable insights for understanding the computational limitations of agentic LLMs. The computational expressiveness of LLMs is fundamentally linked with the ability to solve complex reasoning problems (Wei et al., 2022a). While current models perform well on simple tasks such as navigation and object manipulation, they struggle with more complex tasks that could require non-trivial and general-purpose computation, for example, building a shelter or developing combat strategies. This could be due to the models' inability to retrieve relevant computational circuits (Olah et al., 2020), limitations to inference-time budget (Snell et al., 2024), or representational expressivity. This raises important questions about the scope of effectively solvable tasks for LLMs and VLMs, which is dependent on factors such as model depth, context size, and the distribution shift between pre-training and downstream tasks. Further research is needed to understand the underlying causes of these limitations and to develop strategies for overcoming them, such as adaptive simulation of computational circuits during runtime.

## 7    CONCLUSION

We introduce BALROG, a novel benchmark designed to assess the agentic capabilities of LLMs and VLMs across a diverse set of challenging, long-horizon tasks. Through easily reproducible evaluation protocols, BALROG reveals critical shortcomings in current models, particularly in areas such as vision-based decision-making and long-term planning, identifying clear gaps between model performance and human-level capabilities. These deficiencies, uncovered through our qualitative analysis, reflect the challenges faced in real-world scenarios, underscoring the practical relevance of our benchmark for agentic applications. Our evaluation framework leverages fast, procedurally generated environments, ensuring rigorous and fair comparisons by preventing test-set leakage, a common issue in other benchmarks. We believe that BALROG will serve as a critical tool for supporting and advancing research towards autonomous LLM agents.

## ETHICS STATEMENT

This work provides a benchmark for the agentic capabilities of LLMs. We believe that experimentation in simulated environments, where the behavior of the agents is easy to interpret, is crucial for building safe agentic systems. It is important to address questions on how to ensure that the agent's behavior is well aligned with human intentions.

## REPRODUCIBILITY STATEMENT

We strive to make all experiments in this paper fully reproducible. We share the codebase for evaluation, which is available in the supplementary materials. We describe the full descriptions of the evaluation schemes of the specific environments in Appendices A to F.

## ACKNOWLEDGMENTS

We thank the Gemini Academic Program and Divy Thakkar for their support. We further thank Roberta Raileanu, Pierluca d'Oro, Mikayel Samvelyan, Sam Devlin, Eric Hambro, and Heinrich Küttler for the insightful discussions.

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

## A    BABY AI

BabyAI (Chevalier-Boisvert et al., 2019) is a research platform designed to study grounded language learning and instruction following in artificial agents. It consists of a suite of 2D grid world environments with increasing levels of complexity. In these environments, an agent navigates through rooms and interacts with various objects like doors, keys, balls, and boxes of different colors. The agent receives natural language instructions, called "missions", which describe tasks it needs to complete, such as picking up specific objects or navigating to certain locations. Many existing works on decision-making have studied model performance on this environment (Reed et al., 2022; Li et al., 2022). We use it as a historically relevant environment that we expect to be relatively easy to solve.

### A.1    BABYAI-TEXT

We evaluate the agents on 5 tasks introduced in BabyAI-Text (Carta et al., 2023), which provides a description of each observation instead of a symbolic representation. A textual description consists of a list of template descriptions with the following structure:

- "You see a `<object> <location>`" if the object is a key, a ball, a box or a wall.

- "You see a(n) open/closed door `<location>`", if the agent sees a door.

- "You carry a `<object>`", if the agent carries an object.

### A.2    BABYAI RESULTS

We provide BabyAI results for LLM and VLM mode in Tables 4 and 5. Errors are computed with 25 seeds for each of the 5 tasks of BabyAI. GPT-4o leads, closely followed by Llama 3.1 70B. When vision is added to the observation, GPT4o all models performance decrease, except for Gemini-1.5-Pro, whose performance remains stable.

Table 4: LLM Performance on babyai

| Model | Average Progress (%) |
|---|---|
| gpt-4o | $77.60 \pm 3.73$ |
| llama-3.1-70b-it | $73.20 \pm 3.96$ |
| llama-3.2-90b-it | $72.00 \pm 6.35$ |
| claude-3.5-sonnet | $68.00 \pm 6.60$ |
| gemini-1.5-pro | $58.40 \pm 4.41$ |
| claude-3.5-haiku | $52.00 \pm 7.07$ |
| gpt-4o-mini | $50.40 \pm 4.47$ |
| gemini-1.5-flash | $50.00 \pm 7.07$ |
| llama-3.2-11b-it | $50.00 \pm 7.07$ |
| llama-3.1-8b-it | $36.00 \pm 6.79$ |
| llama-3.2-3b-it | $20.00 \pm 5.66$ |
| llama-3.2-1b-it | $8.00 \pm 3.84$ |

Table 5: VLM Performance on babyai

| Model | Average Progress (%) |
|---|---|
| claude-3.5-sonnet | $82.00 \pm 5.43$ |
| llama-3.2-90b-it | $66.00 \pm 6.70$ |
| gpt-4o | $62.00 \pm 4.34$ |
| gemini-1.5-pro | $58.40 \pm 4.41$ |
| gemini-1.5-flash | $43.20 \pm 4.43$ |
| gpt-4o-mini | $38.00 \pm 4.34$ |
| llama-3.2-11b-it | $18.00 \pm 5.43$ |

### A.3    OBSERVATIONS

Example of instruction prompt and observation for BabyAI

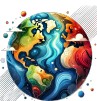

You are an agent playing a simple navigation game. Your goal is to open the yellow door. The following are the possible actions you can take in the game, followed by a short description of each action:

turn left: turn to the left,
turn right: turn to the right,
go forward: take one step forward,
pick up: pick up the object below you,
drop: drop the object that you are holding,
toggle: manipulate the object in front of you.

In a moment I will present you an observation.

Tips:
- Once the desired object you want to interact or pickup in front of you, you can use the 'toggle' action to interact with it.
- It doesn't make sense to repeat the same action over and over if the observation doesn't change.

PLAY!

Current Observation:

a wall 5 steps forward
a wall 2 steps left
a red key 1 step right and 1 step forward
a blue key 1 step right
a yellow key 2 steps right and 3 steps forward
a green key 2 steps right and 1 step forward
a red box 2 steps right
a blue ball 3 steps right and 4 steps forward
a blue box 3 steps right and 1 step forward
a blue box 3 steps right

Image observation provided.

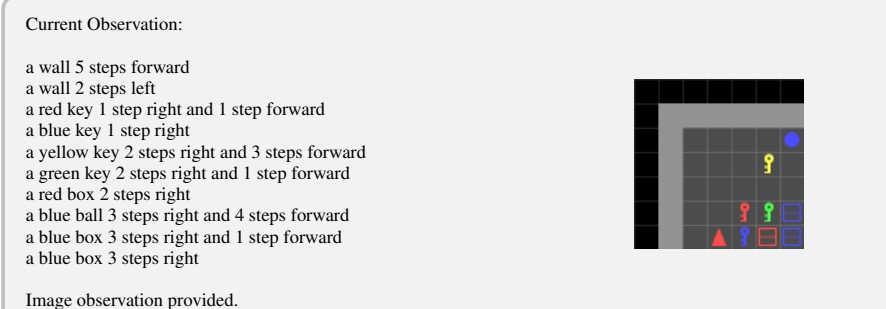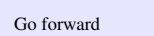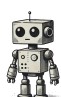

Go forward

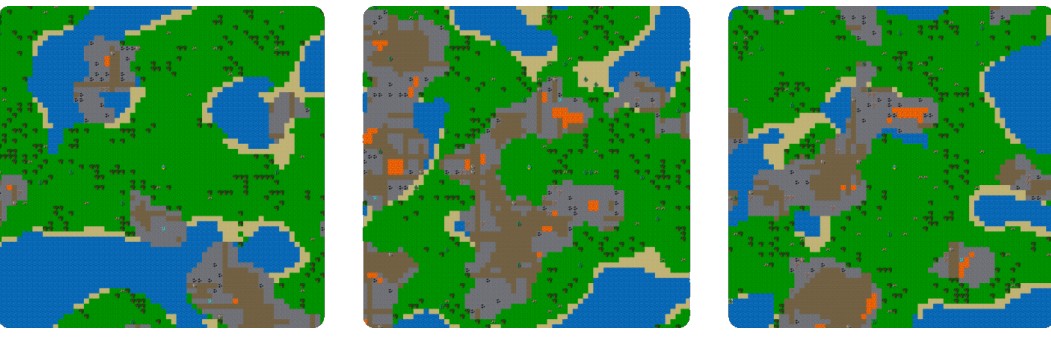

## B CRAFTER

Crafter (Hafner, 2021) is an open-source 2D survival game designed specifically for research on strong generalization, deep exploration, and long-term reasoning in reinforcement learning. It is a Minecraft-inspired, procedurally generated environment that combines resource gathering, crafting, and combat elements. Additionally, the game includes a comprehensive set of tasks and achievements, enabling researchers to evaluate agent performance across multiple objectives and time scales. To enable interaction with language models we use the same language wrapper as proposed in Wu et al. (2023).

Figure 3: Crafter's examples of unique procedurally generated maps.

## B.1 CRAFTER RESULTS

We provide Crafter results for LLM and VLM format in Tables 6 and 7, standard errors are computed using 10 seeds. GPT4o leads in language-only mode, and Gemini-1.5-Pro leads in vision-language mode. Surprisingly, Llama 3.2 90B performance decreases very sharply when images are added, getting worse average progress than its smaller 11B model.

Table 6: LLM Performance on crafter

| Model | Average Progress (%) |
|---|---|
| gpt-4o | $33.10 \pm 2.32$ |
| claude-3.5-sonnet | $32.73 \pm 3.20$ |
| llama-3.2-90b-it | $31.68 \pm 1.36$ |
| llama-3.1-70b-it | $31.21 \pm 2.68$ |
| gemini-1.5-pro | $30.21 \pm 2.86$ |
| claude-3.5-haiku | $26.36 \pm 2.79$ |
| llama-3.2-11b-it | $26.19 \pm 3.29$ |
| llama-3.1-8b-it | $25.45 \pm 3.23$ |
| gemini-1.5-flash | $20.00 \pm 0.74$ |
| llama-3.2-3b-it | $17.27 \pm 2.79$ |
| gpt-4o-mini | $15.90 \pm 2.05$ |
| llama-3.2-1b-it | $12.73 \pm 1.91$ |

Table 7: VLM Performance on crafter

| Model | Average Progress (%) |
|---|---|
| claude-3.5-sonnet | $37.27 \pm 3.14$ |
| gemini-1.5-pro | $33.50 \pm 2.07$ |
| gpt-4o | $26.81 \pm 3.74$ |
| gemini-1.5-flash | $20.70 \pm 4.42$ |
| gpt-4o-mini | $19.91 \pm 3.13$ |
| llama-3.2-11b-it | $15.91 \pm 1.16$ |
| llama-3.2-90b-it | $14.54 \pm 1.80$ |

## B.2 OBSERVATIONS

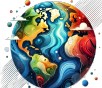

You are an agent playing Crafter. The following are the only valid actions you can take in the game, followed by a short description of each action:

Noop: do nothing,
Move West: move west on flat ground,
Move East: move east on flat ground,
Move North: move north on flat ground,
Move South: move south on flat ground,
Do: Multiuse action to collect material, drink from lake and hit creature in front,
Sleep: sleep when energy level is below maximum,
Place Stone: place a stone in front,
Place Table: place a table,
Place Furnace: place a furnace,
Place Plant: place a plant,
Make Wood Pickaxe: craft a wood pickaxe with a nearby table and wood in inventory,
Make Stone Pickaxe: craft a stone pickaxe with a nearby table, wood, and stone in inventory,
Make Iron Pickaxe: craft an iron pickaxe with a nearby table and furnace, wood, coal, and iron in inventory,
Make Wood Sword: craft a wood sword with a nearby table and wood in inventory,
Make Stone Sword: craft a stone sword with a nearby table, wood, and stone in inventory,
Make Iron Sword: craft an iron sword with a nearby table and furnace, wood, coal, and iron in inventory.

These are the game achievements you can get:

1. Collect Wood
2. Place Table
3. Eat Cow
4. Collect Sampling
5. Collect Drink
6. Make Wood Pickaxe
7. Make Wood Sword
8. Place Plant
9. Defeat Zombie
10. Collect Stone
11. Place Stone
12. Eat Plant
13. Defeat Skeleton
14. Make Stone Pickaxe
15. Make Stone Sword
16. Wake Up
17. Place Furnace
18. Collect Coal
19. Collect Iron
20. Make Iron Pickaxe
21. Make Iron Sword
22. Collect Diamond

In a moment I will present a history of actions and observations from the game. Your goal is to get as far as possible by completing all the achievements.
PLAY!

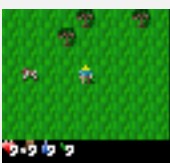

Current Observation:

Your status:
- health: 9/9
- food: 9/9
- drink: 9/9
- energy: 9/9

You have nothing in your inventory.

You see:
- grass 1 steps to your west
- tree 3 steps to your north-west
- cow 3 steps to your west

You face grass at your front.

Image observation provided.

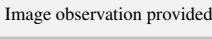

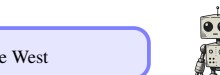

Move West

### B.3 ANALYSIS OF VLM PERFORMANCE IN THE CRAFTER ENVIRONMENT

To investigate the agentic capabilities of VLMs in visually complex environments, we focused on the Crafter environment, a simplified 2D version of Minecraft. The environment uses tiled 2D textures to render scenes, which may be out-of-distribution for many VLMs. To address this potential limitation, we introduced an augmented version of the environment where scenes are rendered in 3D using Minecraft's 3D models and textures. This approach leverages data types more likely to be present in the training datasets of VLMs, given the abundance of interleaved text and images about Minecraft on the web.

**Results** Surprisingly, the switch to 3D rendering did not improve VLM performance. In some cases, performance on the 3D-rendered environment was slightly worse than on the original 2D textures. This result challenges the hypothesis that familiarity with Minecraft-like 3D data in training datasets would lead to better performance. Instead, it suggests that VLMs may struggle with agentic tasks for reasons unrelated to the type of visual texture or rendering style.

**Discussion** Our findings indicate that current VLM models may not be well-suited for complex vision-based reasoning tasks, even when provided with familiar visual contexts. The lack of improvement with 3D rendering suggests that factors other than texture familiarity, such as limitations in spatial reasoning or task-specific adaptation, may play a more significant role in their underperformance. While these preliminary results are insightful, further experimentation is necessary to draw stronger conclusions.

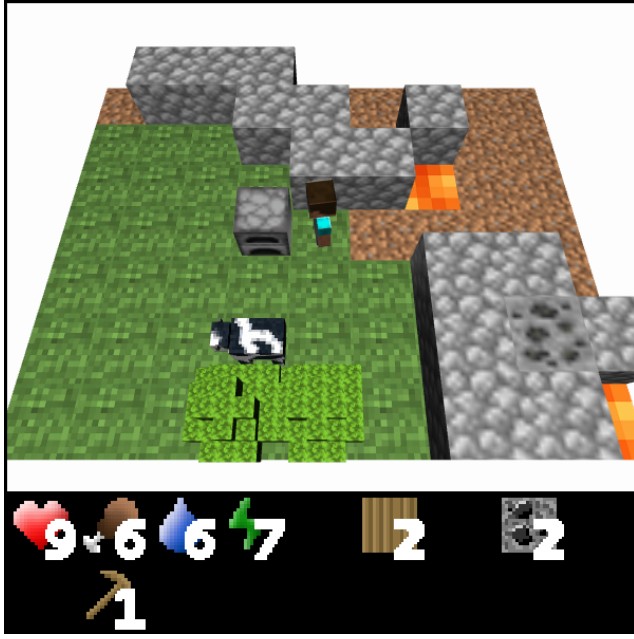

Figure 4: Crafter's example of 3d scene visualization with Minecraft 3d models and textures.

Table 8: Average Progress of VLMs in Crafter Across Rendering Types

| Model | 3d textures(%) | 2d textures(%) |
|---|---|---|
| claude-3.5-sonnet | $25.45 \pm 4.18$ | $37.27 \pm 3.14$ |
| gemini-1.5-pro | $30.45 \pm 3.08$ | $33.50 \pm 2.07$ |
| gpt-4o | $24.55 \pm 3.65$ | $26.81 \pm 3.74$ |
| gemini-1.5-flash | $19.09 \pm 2.78$ | $20.70 \pm 4.43$ |
| gpt-4o-mini | $17.27 \pm 2.38$ | $19.91 \pm 3.13$ |
| llama-3.2-11B-it | $15.90 \pm 1.15$ | $23.63 \pm 1.48$ |
| llama-3.2-90B-it | $12.27 \pm 1.44$ | $10.00 \pm 1.13$ |

# C TEXTWORLD

TextWorld (Côté et al., 2019) is a text-based game environment developed by Microsoft Research that allows for the creation and customization of interactive fiction games. In our experiments, we utilize three specific games from the TextWorld domain: "Treasure Hunter", "The Cooking Game", and "Coin Collector". Each task can be generated with different levels of difficulty by changing number of rooms, enabling obstacles and including distractor rooms. We use the generation rules introduced in Lu et al. (2024).

## C.1 TREASURE HUNTER

In Treasure Hunter, we create a challenging maze-like environment with 20 rooms. The game is set to the maximum difficulty level of 30, introducing locked doors and containers that must be manipulated to locate the target object. To increase complexity, we remove the solution description and filter out tasks that can be solved optimally in 20 steps or fewer. This setup requires the agent to navigate a complex space, interact with various objects, and devise strategies to overcome obstacles in its quest to find the treasure.

## C.2 THE COOKING GAME

The Cooking Game presents a culinary challenge set across 13 rooms. We maximize the complexity by including up to 5 ingredients and enabling all additional challenging options. The agent must navigate through doors, process food items using tools like knives, and cook ingredients using various methods such as grilling, frying, and roasting. This game tests the agent's ability to plan and execute multi-step processes in a dynamic environment, simulating the complexities of real-world cooking tasks.

## C.3 COIN COLLECTOR

Coin Collector features an expansive environment with 40 rooms, including potential distractor rooms to increase navigation difficulty. Similar to Treasure Hunter, we remove the solution description to enhance the challenge. The optimal path from the agent's starting point to the target is set to 20 steps, requiring efficient exploration and decision-making. This game tests the agent's ability to navigate large spaces, avoid distractions, and efficiently reach its goal in a complex, maze-like structure.

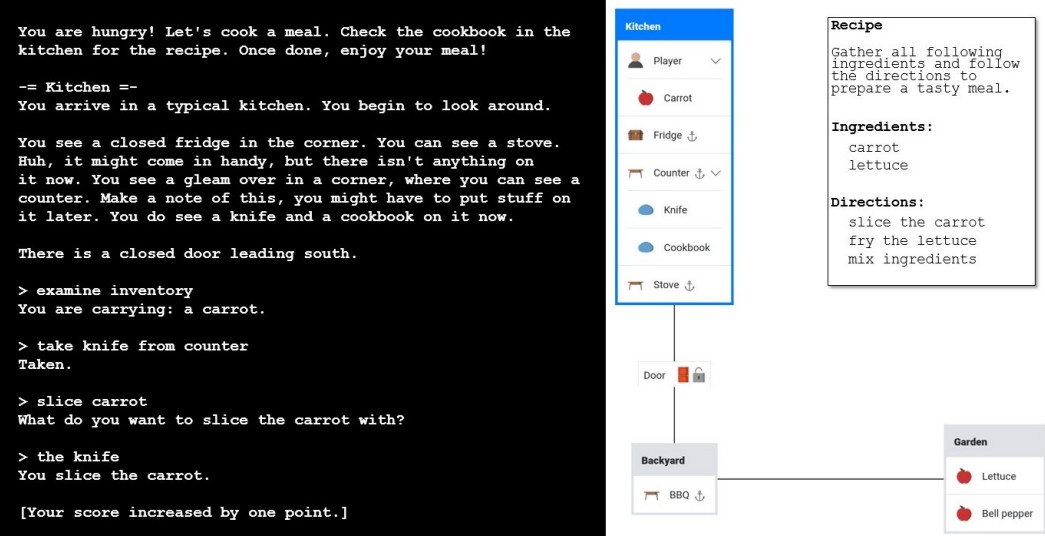

Figure 5: TextWorld interface along with visualization.

## C.4 TEXTWORLD RESULTS

In Table **??**, we provide results for TextWorld. Standard errors are computed using 20 seeds for each of the 3 tasks. GPT-4o once again leads, obtaining more than twice the average progression of its closest competitor Llama 3.1 70B. The coin collector task was by far the most challenging, with GPT-4o managing to solve it only once out of 20 attempts. Gemini models' APIs often failed to return completions on textworld, flagging the inputs as "unsafe", despite there being absolutely no real safety concerns in the textworld gameplays. This made it impossible to complete a full round of evaluation on the Gemini models, thus we marked them as 0% progression.

Table 9: LLM Performance on textworld

| Model | Average Progress (%) |
|---|---|
| claude-3.5-sonnet | $42.06 \pm 5.41$ |
| gpt-4o | $39.31 \pm 5.24$ |
| claude-3.5-haiku | $18.04 \pm 5.81$ |
| llama-3.1-70b-it | $15.00 \pm 4.61$ |
| gpt-4o-mini | $12.25 \pm 3.55$ |
| llama-3.2-90b-it | $11.18 \pm 2.98$ |
| llama-3.2-11b-it | $6.66 \pm 2.17$ |
| llama-3.1-8b-it | $6.08 \pm 2.41$ |
| llama-3.2-3b-it | $3.53 \pm 1.06$ |
| llama-3.2-1b-it | $3.33 \pm 0.91$ |
| gemini-1.5-flash | $0.00 \pm 0.00$ |
| gemini-1.5-pro | $0.00 \pm 0.00$ |

## C.5 OBSERVATIONS

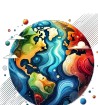

You are an agent playing TextWorld, a text-based adventure game where you navigate through different rooms, interact with objects, and solve puzzles.

Your goal is to first find the recipe, find and prepare food according to the recipe, and finally prepare and eat the meal.

Here are the available commands:

look: describe the current room
goal: print the goal of this game
inventory: print player's inventory
go dir : move the player north, east, south or west. You can only go to directions indicated with an exit or a door.
examine ...: examine something more closely
eat ...: eat edible food
open ...: open a door or a container. You need to open a closed door before you can go through it.
drop ...: drop an object onto the floor
take ...: take an object that is visible
put ... on ...: place an object on a supporter
take ... from ...: take an object from a container or a supporter
insert ... into ...: place an object into a container
lock ... with ...: lock a door or a container with a key
unlock ... with ...: unlock a door or a container with a key
cook ... with ...: cook cookable food with something providing heat
slice ... with ...: slice cuttable food with something sharp
chop ... with ...: chop cuttable food with something sharp
dice ... with ...: dice cuttable food with something sharp
prepare meal: combine ingredients from inventory into a meal.

You can only prepare meals in the Kitchen.
- You can examine the cookbook to see the recipe when it is visible.
- The BBQ is for grilling things, the stove is for frying things, the oven is for roasting things. Cooking ingredients in the wrong way will lead to a failure of the game.
- Once you have got processed ingredients and the appropriate cooking tool ready, cook all of them according to the recipe.
- There are two conditions to correctly cook something (grill/fry/roast):
a) the ingredient you want to cook is in your inventory and
b) there is a suitable cooking tool in the room, and then use 'cook . . . with . . . ' command.
- When you need to chop/slice/dice ingredients, you need to take the knife and the ingredient in your inventory and then 'slice/chop/dice ... with knife'
- Make sure to first process the food (chop/slice/dice) before you try to cook them.
- When you have all the ingredients (that got processed or cooked according to the menu), you can 'prepare meal' in the kitchen and then 'eat meal' to win the game.
- The ingredients should EXACTLY match the color in the recipe, but if the recipe doesn't specify color, any color would be fine. When you 'take ... with ...', use the EXACT name you see.
- You don't need to examine the container/supporter (e.g. toolbox) when it says something like "there isn't a thing on it"/"has nothing on it"

You have 80 steps to complete the task. Restarting is forbidden.

PLAY!

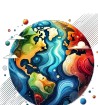

-= Street =-
You find yourself in a street. An usual kind of place.

There is a closed sliding door leading north. There is an exit to the south.

-= Street =-0/1

open sliding door 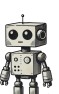

## D BABA IS AI

Baba Is AI is a benchmark environment based on the puzzle game "Baba Is You". In this gridworld game, players interact with various objects and textual rule blocks to achieve specific goals. The unique aspect of Baba Is AI is that the rules of the game can be manipulated and rearranged by the player, creating a dynamic environment where agents must identify relevant objects and rules and then manipulate them to change or create new rules to succeed. This benchmark allows researchers to explore a broader notion of generalization compared to current benchmarks, as it requires agents

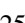

to not only learn and follow the rules but also to combine previously seen rules in novel ways. Agents are tested on 40 different puzzle levels.

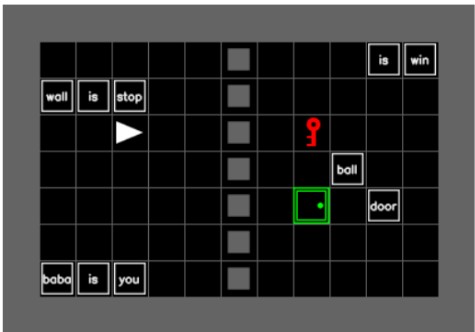

Figure 6: One of the Baba Is AI puzzles, where the agent has to break the "wall is stop" rule, create new rule "door is win" and go to green door to solve the task.

### D.1  BABA IS AI LANGUAGE WRAPPER

To enable interaction with language models, we made a custom language wrapper for Baba Is AI. It constructs language observation from active rules and creates a description by formatting object positions relative to the player. We don't provide the solution for the agent and don't specify grid boundaries in the text-only experiments.

### D.2  BABA IS AI RESULTS

We provide the Baba Is AI results for LLM and VLM mode in Tables **??** and **??**. Standard errors are computed using 5 seeds for each of the 40 Baba Is AI tasks. Surprisingly, the Llama models lead, with Llama 3.2 90B surpassing GPT-4o by a 10% margin in language-only mode. Once again, when vision is added, model performance suffers, with only Gemini-1.5-Pro remaining stable.

Table 10: LLM Performance on babaisai

| Model | Average Progress (%) |
|---|---|
| llama-3.2-90b-it | $43.90 \pm 3.47$ |
| llama-3.1-70b-it | $40.00 \pm 3.42$ |
| claude-3.5-sonnet | $37.50 \pm 4.42$ |
| gpt-4o | $33.66 \pm 3.30$ |
| gemini-1.5-pro | $32.02 \pm 3.26$ |
| llama-3.1-8b-it | $18.33 \pm 3.53$ |
| llama-3.2-3b-it | $17.50 \pm 3.47$ |
| gpt-4o-mini | $15.60 \pm 2.53$ |
| llama-3.2-11b-it | $15.60 \pm 2.50$ |
| gemini-1.5-flash | $12.80 \pm 2.33$ |
| llama-3.2-1b-it | $10.83 \pm 2.84$ |
| claude-3.5-haiku | $8.33 \pm 2.52$ |

Table 11: VLM Performance on babaisai

| Model | Average Progress (%) |
|---|---|
| claude-3.5-sonnet | $34.45 \pm 4.36$ |
| gemini-1.5-pro | $31.40 \pm 3.24$ |
| llama-3.2-90b-it | $21.90 \pm 2.89$ |
| gpt-4o | $18.62 \pm 2.72$ |
| gpt-4o-mini | $16.41 \pm 2.59$ |
| gemini-1.5-flash | $8.30 \pm 1.92$ |
| llama-3.2-11b-it | $5.76 \pm 1.63$ |

### D.3 OBSERVATIONS

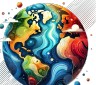

Baba Is You is a puzzle game where you can manipulate the rules of each level. The following are the possible actions you can take in the game, followed by a short description of each action:

idle: wait for one step,
up: take one step up,
right: take one step to the right,
down: take one step down,
left: take one step to the left.

Tips:
- Examine the level carefully, noting all objects and text blocks present.
- Identify the current rules, which are formed by text blocks in the format "[Subject] IS [Property]" (e.g. "BABA IS YOU").
- Consider how you can change or create new rules by moving text blocks around.
- Remember that you can only move objects or text that are not defined as "STOP" or similar immovable properties.
- Your goal is usually to reach an object defined as "WIN", but this can be changed.
- Think creatively about how changing rules can alter the properties and behaviors of objects in unexpected ways.
- If stuck, try breaking apart existing rules or forming completely new ones.
- Sometimes the solution involves making yourself a different object or changing what counts as the win condition.

PLAY!

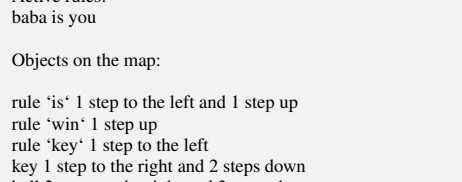

Current Observation:

Active rules:
baba is you

Objects on the map:

rule 'is' 1 step to the left and 1 step up
rule 'win' 1 step up
rule 'key' 1 step to the left
key 1 step to the right and 2 steps down
ball 2 steps to the right and 3 steps down
rule 'baba' 2 step to the left and 4 steps down
rule 'is' 1 step to the left and 4 steps down
rule 'you' 4 steps down
rule 'ball' 2 steps to the right and 4 steps down

Image observation provided.

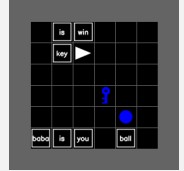

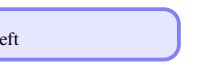

left

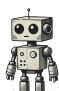

## E  MINIHACK

MiniHack (Samvelyan et al., 2021) is a powerful sandbox framework built on top of the NLE (Küttler et al., 2020) that enables researchers to easily design rich and diverse environments for RL. It provides a flexible platform for creating custom RL tasks ranging from simple grid-world navigation to complex, procedurally generated worlds with intricate game mechanics. The framework allows users to define environments using a human-readable description language or a simple Python interface, giving fine-grained control over environment elements such as terrain, objects, monsters, and traps. MiniHack offers a diverse array of tasks, which can be broadly categorized into three main groups: Navigation Tasks, Skill Acquisition Tasks, and Ported Tasks. To enable interaction with language models, we use NetHack Language Wrapper described in the NetHack Appendix F.

From the MiniHack Navigation Tasks, we picked Maze 9x9, Maze 15x15, Corridor and Corridor-Battle, which challenge the agent to reach the goal position by overcoming various difficulties on their way, such as fighting monsters in corridors and navigating through complex or procedurally generated mazes. These tasks feature a relatively small action space, i.e., movement towards 8 compass directions, and based on the environment, search, kick, open, and eat actions.

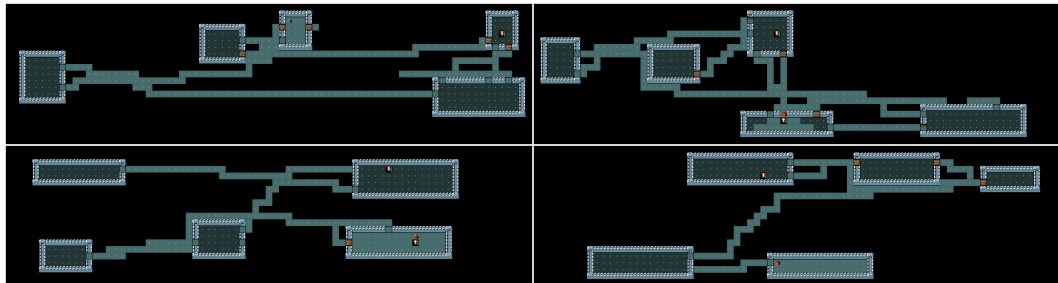

Figure 7: Examples of MiniHack Corridor task.

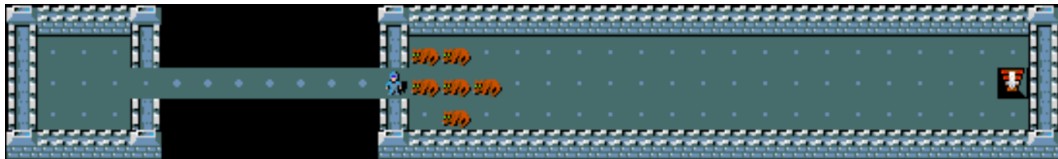

Figure 8: Example of MiniHack CorridorBattle task.

From the MiniHack Skill Acquisition Tasks, we picked Quest (with two different difficulty levels, Easy and Medium), which challenges the agent to use objects found in the environment to cross a lava river (these objects can provide levitation or freezing abilities), fight monsters, navigate through rooms or mazes and towards the end of the quest use a wand of death to defeat a powerful monster guarding the goal location.

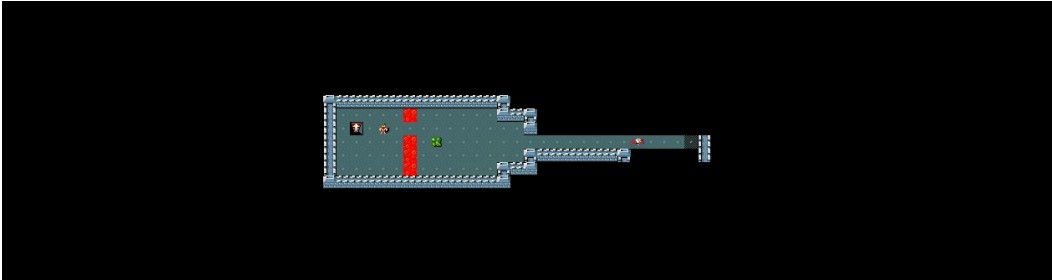

Figure 9: Example of MiniHack Quest Easy task.

We additionally test the agents on MiniHack Boxoban. This family of environments is an adaptation of the Boxoban puzzle game, which itself is inspired by the classic Sokoban. These environments present a challenging puzzle-solving task within the MiniHack framework, leveraging the NetHack game mechanics. The primary goal in MiniHack Boxoban is to push four boulders (MiniHack's equivalent of boxes) onto four designated goal locations, which are represented by fountains. This task requires strategic thinking and planning, as the agent must carefully maneuver the boulders through the environment without getting them stuck in corners or against walls.

We provide MiniHack results for LLM and VLM mode in Tables 12 and 13, standard errors were computed using 5 seeds for each task. Here, GPT-4o and a Gemini-1.5-Pro equal each other both in language-only and vision-language mode, with both models only managing to complete some of the corridor and corridor battle tasks. None of the other models solved any task.

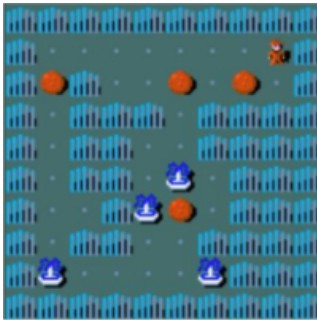

Figure 10: Example of MiniHack Boxoban Hard task.

Table 12: LLM Performance on minihack

| Model | Average Progress (%) |
|---|---|
| claude-3.5-sonnet | 15.00 ± 5.65 |
| claude-3.5-haiku | 10.00 ± 4.74 |
| gpt-4o-mini | 10.00 ± 4.74 |
| gpt-4o | 10.00 ± 4.74 |
| llama-3.1-70b-it | 7.50 ± 4.16 |
| gemini-1.5-flash | 5.00 ± 3.45 |
| gemini-1.5-pro | 5.00 ± 3.45 |
| llama-3.2-1b-it | 5.00 ± 3.45 |
| llama-3.1-8b-it | 5.00 ± 3.45 |
| llama-3.2-90b-it | 5.00 ± 3.44 |
| llama-3.2-3b-it | 2.50 ± 2.47 |
| llama-3.2-11b-it | 2.50 ± 2.47 |

Table 13: VLM Performance on minihack

| Model | Average Progress (%) |
|---|---|
| claude-3.5-sonnet | 22.50 ± 6.60 |
| gpt-4o | 5.00 ± 3.44 |
| gemini-1.5-pro | 5.00 ± 3.44 |
| gpt-4o-mini | 2.50 ± 2.47 |
| gemini-1.5-flash | 2.50 ± 2.47 |
| llama-3.2-11b-it | 2.50 ± 2.46 |
| llama-3.2-90b-it | 2.50 ± 2.47 |

## E.1 OBSERVATIONS

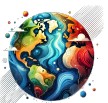

You are an agent playing MiniHack. The following are the possible actions you can take in the game, followed by a short description of each action:
north: move north,
east: move east,
south: move south,
west: move west,
northeast: move northeast,
southeast: move southeast,
southwest: move southwest,
northwest: move northwest,
far north: move far north,
far east: move far east,
far south: move far south,
far west: move far west,
far northeast: move far northeast,
far southeast: move far southeast,
far southwest: move far southwest,
far northwest: move far northwest,
down: go down the stairs,
wait: rest one move while doing nothing,
more: display more of the message,
apply: apply (use) a tool,
close: close an adjacent door,
eat: eat something,
force: force a lock,
kick: kick an enemy or a locked door or chest,
loot: loot a box on the floor,
open: open an adjacent door,
pickup: pick up things at the current location if there are any,
pray: pray to the gods for help,
puton: put on an accessory,
quaff: quaff (drink) something,
search: search for hidden doors and passages,
zap: zap a wand.

In a moment I will present a history of actions and observations from the game.

Your goal is to explore the level, fight monsters, and navigate rooms and mazes to ultimately reach the stairs down.

PLAY!

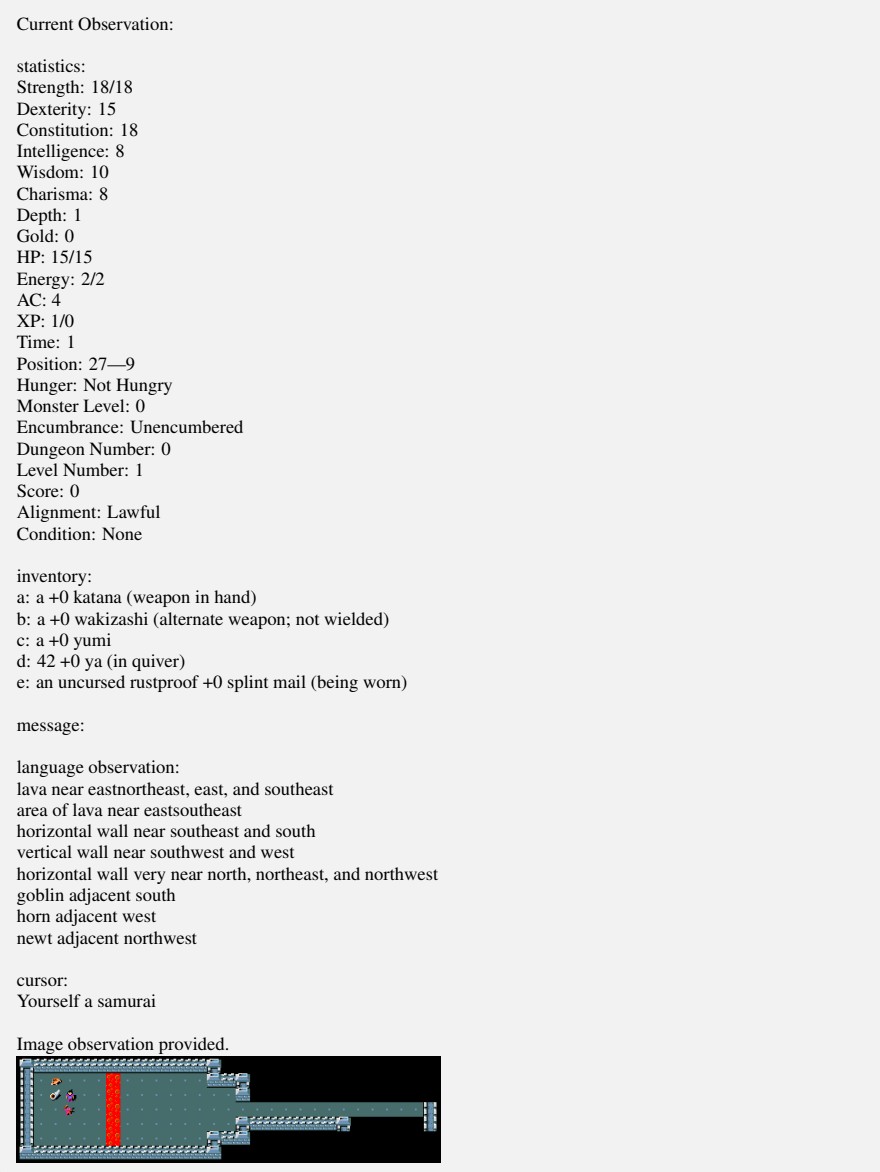

Current Observation:

statistics:
Strength: 18/18
Dexterity: 15
Constitution: 18
Intelligence: 8
Wisdom: 10
Charisma: 8
Depth: 1
Gold: 0
HP: 15/15
Energy: 2/2
AC: 4
XP: 1/0
Time: 1
Position: 27—9
Hunger: Not Hungry
Monster Level: 0
Encumbrance: Unencumbered
Dungeon Number: 0
Level Number: 1
Score: 0
Alignment: Lawful
Condition: None

inventory:
a: a +0 katana (weapon in hand)
b: a +0 wakizashi (alternate weapon; not wielded)
c: a +0 yumi
d: 42 +0 ya (in quiver)
e: an uncursed rustproof +0 splint mail (being worn)

message:

language observation:
lava near eastnortheast, east, and southeast
area of lava near eastsoutheast
horizontal wall near southeast and south
vertical wall near southwest and west
horizontal wall very near north, northeast, and northwest
goblin adjacent south
horn adjacent west
newt adjacent northwest

cursor:
Yourself a samurai

Image observation provided.

south

# F   NETHACK LEARNING ENVIRONMENT

The NetHack Learning Environment (NLE) (Küttler et al., 2020) is a scalable, procedurally generated, stochastic, rich, and challenging environment designed to drive long-term research in RL on problems such as exploration, planning, skill acquisition, and language-conditioned RL. Built around the classic and highly complex terminal roguelike game NetHack, NLE provides a complex and dynamic environment where agents must navigate through procedurally generated dungeons, interact with hundreds of entity types, and learn to overcome various challenges.

The goal of the player is to descend through procedurally generated dungeon levels while killing monsters, solving puzzles, and gathering better equipment in order to retrieve the Amulet of Yendor and finally ascend back to the surface to win the game. NetHack is notoriously challenging, even for human players. Mastering the game can take years even with online resources like the

NetHack Wiki. Success in NetHack demands long-term strategic planning, as a winning game can involve hundreds of thousands of steps, as well as short-term tactics to fight hordes of monsters. Accurate credit assignment is also crucial to understanding which actions contributed to success or failure. NetHack has already been used extensively as a testbed for RL agents (Wołczyk et al., 2024; Piterbarg et al., 2024; Hambro et al., 2022b); tabula-rasa RL agents particularly struggle due to sparse reward, complex credit assignment, extremely long-time-horizon, and high stochasticity of the game. The current state-of-the-art agent still remains a hand-coded symbolic policy (Hambro et al., 2022a).

## F.1 NetHack Language Wrapper

The NetHack Language Wrapper (Goodger et al., 2023) is a tool designed to interface with the NLE and MiniHack by translating non-language observations into text-based representations. This wrapper, converts various NLE observations such as `glyphs`, `blstats`, `tty_chars`, `inv_letters`, `inv_strs`, and `tty_cursor` into readable text equivalents. For example, it transforms the visual display of the game environment into a textual description, including details about the surroundings, inventory, and player statistics. The wrapper also supports text-based actions, allowing users to interact with the environment using commands like `wait`, `apply`, and `north`, which are then converted into the discrete actions required by the NLE. This functionality enables easier interaction with the NetHack environment, particularly for language models.

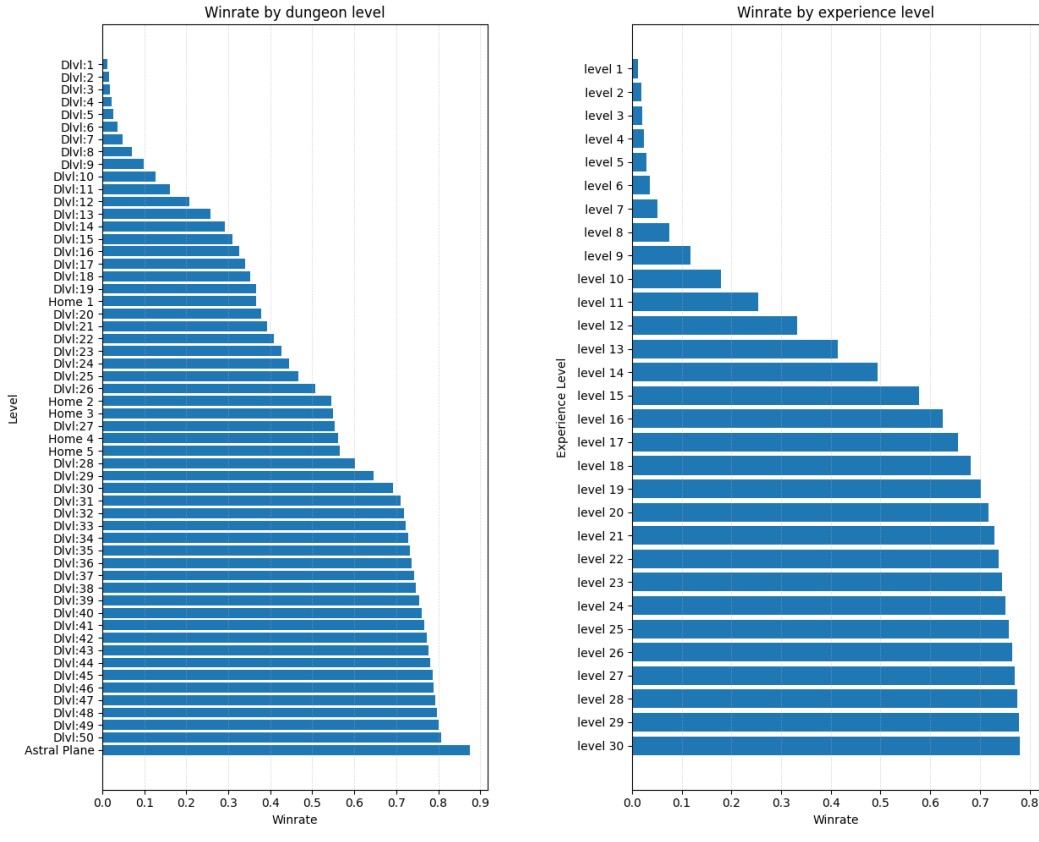

(a) NetHack progression by dungeon level reached    (b) NetHack progression by experience level

Figure 11

## F.2 New NetHack Progression System

NetHack features an in-game scoring system that rewards players for actions such as killing monsters, identifying objects, eating food, collecting gold and items, and ultimately ascending in the

game. However, we argue that this scoring system does not effectively capture true game progression, as players can win the game with scores ranging from a few hundred thousand to several million points. To address this limitation, we developed a novel, data-informed progression metric using a dataset of human-played NetHack games (Hambro et al., 2022b). Specifically, we recorded the dungeon levels and experience levels achieved in each game, as well as whether the game resulted in an ascension. Utilizing these statistics, we constructed a data-centric progression system where each data point represents the probability of a human player winning the game after reaching a specific dungeon level or experience level. The resulting progression curves are presented in Figure 11. For practical purposes, we define Dungeon Level 1 (Dlvl:1) and Experience Level 1 as representing 0% progression, corresponding to the game's starting point, and ascension as 100% progression. The agent's overall progress is thus determined by the highest progression achieved between the dungeon level and experience level attained.

### F.3 NETHACK RESULTS

We provide NetHack results for LLM and VLM mode in Tables 14 and 15. Standard errors are computed using 5 seeds. o1-preview achieves the highest progression out of all the tested models. However, it is still very far from making any significant progression in the game. The best individual run was achieved by Gemini-1.5-Pro vision-language mode, reaching dungeon level 3 and experience level 4.



Table 14: LLM Performance on nle

| Model | Average Progress (%) |
|---|---|
| o1-preview | $1.57 \pm 0.40$ |
| claude-3.5-haiku | $1.16 \pm 0.42$ |
| claude-3.5-sonnet | $0.58 \pm 0.52$ |
| gpt-4o | $0.37 \pm 0.37$ |
| gemini-1.5-pro | $0.37 \pm 0.37$ |
| llama-3.1-70b-it | $0.35 \pm 0.35$ |
| gpt-4o-mini | $0.00 \pm 0.00$ |
| gemini-1.5-flash | $0.00 \pm 0.00$ |
| llama-3.2-1b-it | $0.00 \pm 0.00$ |
| llama-3.2-3b-it | $0.00 \pm 0.00$ |
| llama-3.1-8b-it | $0.00 \pm 0.00$ |
| llama-3.2-11b-it | $0.00 \pm 0.00$ |
| llama-3.2-90b-it | $0.00 \pm 0.00$ |

Table 15: VLM Performance on nle

| Model | Average Progress (%) |
|---|---|
| claude-3.5-sonnet | $1.16 \pm 0.42$ |
| gemini-1.5-pro | $0.48 \pm 0.48$ |
| gpt-4o | $0.37 \pm 0.37$ |
| gpt-4o-mini | $0.00 \pm 0.00$ |
| gemini-1.5-flash | $0.00 \pm 0.00$ |
| llama-3.2-11b-it | $0.00 \pm 0.00$ |
| llama-3.2-90b-it | $0.00 \pm 0.00$ |



### F.4 OBSERVATION

Despite having a language wrapper that describes its observations (Goodger et al., 2023), NetHack is not meant to be played with language only, thus we provided the ASCII map in language mode and the RGB tiles map in vision-language mode. In the LLM context, we only keep information that is important to be kept in the long term, i.e., the game message and language observation. Agent stats and inventory are only needed in the current step, so we do not keep them in the context. This is done also to prevent the context length of NetHack to explode out of control.

You are an agent playing NetHack.
The following are the possible actions you can take in the game, followed by a short description of each action:

north: move north,
east: move east,
south: move south,
west: move west,
northeast: move northeast,
southeast: move southeast,
southwest: move southwest,
northwest: move northwest,
far north: move far north,
far east: move far east,
far south: move far south,
far west: move far west,
far northeast: move far northeast,
far southeast: move far southeast,
far southwest: move far southwest,
far northwest: move far northwest,
up: go up a staircase,
down: go down a staircase (tip: you can only go down if you are standing on the stairs),
wait: rest one move while doing nothing,
more: display more of the message (tip: ONLY ever use when current message ends with –More–),
annotate: leave a note about the level,
apply: apply (use) a tool,
call: name a monster or object, or add an annotation,
cast: cast a spell,
close: close an adjacent door,
open: open an adjacent door,
dip: dip an object into something,
drop: drop an item,
droptype: drop specific item types (specify in the next prompt),
eat: eat something (tip: replenish food when hungry),
esc: exit menu or message,
engrave: engrave writing on the floor (tip: Elbereth),
enhance: advance or check weapons skills,
fire: fire ammunition from quiver,
fight: fight a monster (even if you only guess one is there),
force: force a lock,
[...]
read: read a scroll or spellbook,
remove: remove an accessory,
rub: rub a lamp or a stone,
search: search for hidden doors and passages,
swap: swap wielded and secondary weapons,
takeoff: take off one piece of armor,
takeoffall: take off all armor,
teleport: teleport to another level (if you have the ability),
throw: throw something (e.g. a dagger or dart),
travel: travel to a specific location on the map (tip: in the next action, specify <or for stairs, { for fountain, and _ for altar),
twoweapon: toggle two-weapon combat,
untrap: untrap something,
wear: wear a piece of armor,
wield: wield a weapon,
wipe: wipe off your face,
zap: zap a wand,

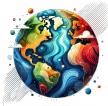

Tips:
- When the message asks for a completion, such as: "What do you want to eat? [d or ?*]", you should respond with a single character corresponding to the item you want to eat/use.
- For example, "What do you want to eat? [dgh or ?*]" -¿ Possible answers are "d", "g", or "h" to eat the associated food.
- When the message asks for a direction, such as: "In what direction?" you should respond with a direction.
- When the message has –More– at the end, your next action should be "more" to see the rest of the message.
- Explore the environment to find the stairs down to the next level.
- Always carefully read the last message to understand the current state of the game and decide your next action accordingly.
- If you keep moving in the same direction, you will eventually hit a wall and stop moving. Your message might be: "It's solid stone", or "It's a wall". Change your action to move in another direction to continue exploring the environment.
- Read the language observation carefully and look at ascii map or image observation provided to decide the next action to take and where to move next.
- You can attack monsters by moving into them.

In a moment I will present a history of actions and observations from the game.
Your goal is to get as far as possible in the game.

PLAY!

Current Observation:

statistics:
Strength: 14/14
Dexterity: 9

Constitution: 11
Intelligence: 8
Wisdom: 15
Charisma: 18
Depth: 1
Gold: 0
HP: 16/16
Energy: 6/6
AC: 3
XP: 1/0
Time: 1
Position: 54—14
Hunger: Not Hungry
Monster Level: 0
Encumbrance: Unencumbered
Dungeon Number: 0
Level Number: 1
Score: 0
Alignment: Lawful
Condition: None

inventory:
a: a +1 long sword (weapon in hand)
b: a +1 lance (alternate weapon; not wielded)
c: an uncursed +1 ring mail (being worn)
d: an uncursed +0 helmet (being worn)
e: an uncursed +0 small shield (being worn)
f: an uncursed +0 pair of leather gloves (being worn)
g: 11 uncursed apples
h: 10 uncursed carrots

message:
Salutations Agent, welcome to NetHack! You are a lawful female human Knight.

language observation:
vertical closed door far east
dark area far east
scroll labeled THARR near eastsoutheast
horizontal wall near southeast and south
dark area near west
vertical wall very near southwest
vertical open door very near west
horizontal wall adjacent north, northeast, and northwest
tame pony adjacent southeast

cursor:
Yourself a knight

Image observation provided.

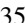

west

## F.5 LLMs possess extensive NetHack knowledge

Agents need extensive knowledge to play NetHack effectively. Skilled players leverage their understanding of the game, acquired through hours of gameplay, while new players often rely on written

resources online, such as the NetHack Wiki. We use a questionnaire to test if LLMs possess some of this knowledge and assess if agents can apply it. The questions are as follows:

1. Are there any consequences of eating rotten food in NetHack? **(Answer: It might lead to blindness, hallucination or even instant death.)**

2. Are there any consequences of eating too much food in NetHack? **(Answer: The player character might choke on the food and die.)**

3. What happens when you ascend the stairs on level 1 of NetHack? **(Answer: It leads to instant game over.)**

4. Is it a good idea to use an invisibility wand on a monster in NetHack? **(Answer: No, as it will be harder to fight them.**)

5. Is it a good idea to pray to an angered god in NetHack? **(Answer: No, the god will punish the player.**)

These questions test a fundamental understanding of the game mechanics, particularly focusing on behaviors that new players may mistakenly attempt and should be avoided.

We summarize each LLM's responses in the tables below. For each question, we assess whether the response is accurate, whether the conclusion is correct (i.e., the LLM recognizes the need to avoid such behavior), and whether the agent successfully avoids these mistakes.

| LLM | | Q1 | Q2 | Q3 | Q4 | Q5 |
|---|---|---|---|---|---|---|
| | Correct | ✔ | ✔ | ∼ | ✔ | ✔ |
| GPT 4o | Conclusion | ✔ | ✔ | ✔ | ✔ | ✔ |
| | Behaviour | ✗ | ✔ | ✗ | N/A | ✔ |
| | Correct | ∼ | ✗ | ✔ | ✗ | ✔ |
| GPT 4o-mini | Conclusion | ✔ | ✔ | ✔ | ✔ | ✔ |
| | Behaviour | ✗ | ✔ | ✔ | N/A | N/A |
| | Correct | ✗ | ✗ | ✗ | ✗ | ✔ |
| Gemini 1.5-flash | Conclusion | ✔ | ✗ | ✗ | ✗ | ✔ |
| | Behaviour | ✔ | ✔ | ✗ | N/A | N/A |
| | Correct | ✔ | ∼ | ✗ | ✔ | ✔ |
| Gemini 1.5-pro | Conclusion | ✔ | ✔ | ✗ | ✔ | ✔ |
| | Behaviour | ✔ | ✔ | ✗ | N/A | N/A |
| | Correct | ✔ | ✗ | ✔ | ✗ | ✔ |
| Llama 3.1 70B Instruct | Conclusion | ✔ | ✗ | ✗ | ✔ | ✔ |
| | Behaviour | ✗ | ✗ | ✗ | ✗ | ✗ |
| | Correct | ✗ | ✗ | ✗ | ✗ | ✔ |
| Llama 3.2 11B Instruct | Conclusion | ✔ | ✗ | ✗ | ✔ | ✔ |
| | Behaviour | ✗ | ✗ | ✗ | N/A | N/A |
| | Correct | ✔ | ∼ | ✔ | ✗ | ✔ |
| Llama 3.2 90B Instruct | Conclusion | ✔ | ✔ | ✔ | ✔ | ✔ |
| | Behaviour | ✗ | ✔ | ✗ | N/A | N/A |

Table 16: Comparison of each LLMs (ability to apply) knowledge in Nethack. We manually grade the responses to each question based on the **correctness** of the response given (i.e. does the response match information from the Nethack wiki), the correctness of their **conclusion** (i.e. does the LLM correctly identify that such behaviour should be avoided), and whether an LLM agent's **behaviour** during evaluation is consistent with the ground truth (i.e. does the agent successfully avoid the behaviours indicated in the questions). For answers that are partially correct, we award a ∼. We record behaviour as N/A when the agent does not encounter scenarios where knowledge of the corresponding question should be applied.

We observe that while generally the LLMs understand to avoid common mistakes, regardless of whether their reasoning is completely correct, they still struggle to consistently exploit that knowledge. Agents will often consume rotten food and prematurely exit the game by ascending the steps on the first level. This illustrates a gap between LLM agents ability to exploit knowledge in practice.

