# OpenReview forum: "BALROG: Benchmarking Agentic LLM and VLM Reasoning On Games"
_ICLR.cc/2025/Conference — ICLR 2025 Poster_

### Official Review · Reviewer_SuiJ · 2024-10-21

**Soundness:** 3
**Presentation:** 3
**Contribution:** 3
**Rating:** 6
**Confidence:** 3

**Summary:**

This paper presents a benchmark to evaluate LLMs' capabilities in video games. Six video games are included. One metric that captures how close the LLMs are to completing a task is proposed for evaluation. State-of-the-art LLMs are used for evaluation. The paper presents findings from quantitative and qualitative experiments.

**Strengths:**

1. The presentation of this benchmark is timely.
2. The paper presents problems about the knowing-doing gap, which might spark more research.

**Weaknesses:**

My major concern is the comprehensiveness of the proposed benchmark. Current VLMs or MLLMs are usually trained on image-text pairs in the real world. The appearance of the involved video games might differ largely from the training data. It is not very surprising, at least to the reviewer, that providing game images leads to worse performance. More scenarios, especially general ones, should be involved to support the conclusion. This can also further reflect the overall capability of MLLMs.

**Questions:**

1. What does the "reasoning" refer to in L72 "action = self.llm_cilent(reasoning)"?

---

> ### Author Response · Authors · 2024-11-18
>
> We thank the reviewer for their thoughtful feedback and for acknowledging the timeliness of our benchmark, and the potential impact of having identified the “knowing-doing” gap.
>
> We address your concern and question below:
>
> ### **[Q1] Comprehensiveness of the benchmark and VLM results**
> We appreciate the reviewer’s concern regarding the comprehensiveness of the benchmark and its evaluation of VLMs. While it is true that the visuals in video games may be somewhat out-of-distribution (OOD) compared to the training data of current VLMs, many models have been exposed to a variety of multimodal datasets, potentially including content related to video games. This makes their underperformance in these scenarios an interesting result worth further investigation, as it points to challenges in leveraging visual information for decision-making in dynamic environments.
>
> AgentQuest, as a benchmark, aims to provide a rigorous framework for evaluating models in complex, long-horizon tasks. The fact that VLMs struggle in certain tasks is not a limitation of the benchmark but rather a reflection of the current state of model capabilities. These results highlight specific gaps in the decision-making abilities of VLMs, particularly in leveraging visual context in dynamic settings—a challenge also shared by standard LLMs. We believe this is precisely the role of a good benchmark: to reveal weaknesses that inspire further research and innovation.
>
> To further address your concern, we conducted additional experiments with claude 3.5 sonnet in VLM mode. Interestingly, this model showed improvements in performance when using visual input compared to its language-only counterpart, suggesting that some VLMs can benefit from visual observations even in challenging OOD scenarios. We include these updated results in our fully anonymous leaderboard website https://agentquestanon.github.io/ and in the revised paper.
>
> ### **[Q2] Clarifying reasoning in line 72**
> We apologize for the confusion. The line action = self.llm_cilent(reasoning) contains an error. Instead of “reasoning” it should be written “history”. We thank the Reviewer for spotting this and we have fixed the issue in the revised version of the paper.
>
> ### **Conclusion**
> We thank the reviewer for their constructive feedback, which has helped us strengthen the comprehensiveness and clarity of our work. We hope the added experiments and clarifications addressed your concerns and hope that based on this you will consider increasing the score for our paper, or point out what still stands in the way for a higher rating.

---

> > ### Author Response · Authors · 2024-11-22
> >
> > Dear Reviewer **SuiJ**
> >
> > We hope this message finds you well.
> >
> > We have carefully addressed all your questions and concerns, conducted additional evaluations of more agentic models, and revised the paper accordingly.
> >
> > As the rebuttal deadline is approaching soon, we kindly ask if you could strengthen support to the paper, or let us know if you have any further questions and we will reply swiftly.
> >
> > Kind regards,
> >
> > The Authors

---

> > > ### Comment · Reviewer_SuiJ · 2024-11-24
> > >
> > > Thanks for the responses. Some of my concerns are addressed. I think long-horizon interactive tasks should not only involve video games, and I will keep my original rating.

---

> ### Author Response · Authors · 2024-11-25
>
> Thank you for your feedback. Games have a long tradition as a strong metric for AI due to their high complexity, long horizon and interactivity. Identifying tasks that are both long-horizon and interactive outside of games that are appropriate for both LLM and VLM evaluation is challenging.
>
> Web-browsing tasks typically involve shorter decision chains compared to games, lacking the deep sequential dependencies that define long-horizon challenges.
>
> While coding benchmarks can be considered somewhat long-horizon and interactive, they lack the dynamic, real-time decision-making and feedback loops that games inherently provide and that we are interested in. Moreover, there are already numerous coding benchmarks available.
>
> We would also not expect a particular benchmark to cover all long-horizon interactive tasks one can imagine. However, if you have specific suggestions or domains in mind, we'd greatly appreciate your input.

---

> > ### Author Response · Authors · 2024-11-28
> >
> > Dear Reviewer, to further address your comments regarding VLM performance and the comprehensiveness of the benchmark, we conducted new experiments using a 3D Minecraft-style rendering of the Crafter environment. The findings are included in our updated shared rebuttal and detailed in Appendix B.3 of the revised paper. We invite you to review these updates and would greatly value your input.

---

### Official Review · Reviewer_HZcx · 2024-11-03

**Soundness:** 2
**Presentation:** 3
**Contribution:** 3
**Rating:** 5
**Confidence:** 4

**Summary:**

This paper introduces a new benchmark designed to assess the agent ability of both LLMs and VLMs, with a particular focus on their performance in complex environments. By incorporating diverse tasks and refined metrics, this benchmark highlights the strengths and limitations of current models, especially in visual decision-making tasks where shortcomings are evident. Evaluation results indicate that while current models perform well on simple tasks, they continue to face significant challenges with complex ones. AgentQuest provides a platform for advancing research in intelligent agents.

**Strengths:**

1. AgentQuest integrates various reinforcement learning environments, covering tasks from simple to highly complex, to comprehensively evaluate the agency capabilities of LLMs and VLMs.

2. Different metrics have been designed to provide quantitative analysis of model performance across multiple dimensions, including interactivity, spatial reasoning, and long-term planning.

3. According to the authors, AgentQuest is convenient for researchers to expand and apply, helping to advance the field of intelligent agents and improve model usability in complex, dynamic environments.

**Weaknesses:**

1. The main issue is that this paper, as a benchmark for evaluating the agent capabilities of models, only assesses base models. Many recent studies focus on enhancing models’ agent capabilities, but this paper does not evaluate these works. While base models are important, they weren’t specifically trained to improve agent abilities. This may be one reason why they perform poorly on many tasks. Evaluating the latest agent-focused methods would better reflect the field's progress.

2. As far as I know, this paper simply combines several existing benchmarks without further integration. While bringing existing benchmarks together to evaluate model performance may be important, it doesn’t qualify as innovative work. The authors need to provide further explanation on this point.

**Questions:**

Creating a comprehensive benchmark to test model agent capabilities is very meaningful; however, the authors' work has certain limitations. It would be helpful if the authors could provide further explanation.

---

> ### Author Response · Authors · 2024-11-18
>
> We thank the reviewer for their thoughtful feedback and for recognizing the strengths of AgentQuest, including its integration of diverse environments and comprehensive evaluation metrics. We appreciate your constructive criticism, which has helped us improve the clarity and scope of our work. Below, we address each point raised.
>
> ### **[Q1] Evaluating more agentic models**
> We appreciate the reviewer highlighting the importance of evaluating models specifically designed to enhance agentic capabilities. AgentQuest establishes a baseline for widely-used instruction-following models in a zero-shot setting, providing the research community with a foundation for future innovation. As a benchmark paper, our primary goal is to enable the research community to evaluate and improve agentic models systematically.
>
> That said, we fully agree that testing more advanced models that have been designed to enhance agentic performance is important. AgentQuest is designed to facilitate this kind of evaluation of fine-tuned models, inference-time strategies, and reinforcement learning approaches, as described in section 2.2. To address your concern, we have expanded our experiments in the revised paper to include additional models, specifically we added Claude 3.5 Sonnet, smaller Llama-3.1/3.2 variants (1B/3B/8B)-it, and more. Claude 3.5 sonnet has been trained on computer interaction [1], which likely enhances its capabilities in agentic tasks like AgentQuest with visual inputs, which may explain its state-of-the-art performance on the benchmark in VLM mode.
>
> | Model               | Progress ± SE         |
> |---------------------|-----------------------|
> | Claude-3.5-Sonnet   | 30.0 ± 2.0            |
> | Qwen2.5-72B-it      | 16.2 ± 1.6            |
> | Llama-3.1-8B-it     | 14.1 ± 1.5            |
> | Qwen2-VL-72B-it     | 12.8 ± 1.6            |
> | Llama-3.2-3B-it     | 8.5 ± 1.1             |
> | Qwen-2.5-7B-it      | 7.8 ± 1.1             |
> | Llama-3.2-1B-it     | 6.3 ± 1.0             |
> | Qwen2-VL-7B-it      | 3.7 ± 0.8             |
>
>
> | Model             | Progress ± SE         |
> |-------------------|-----------------------|
> | Claude-3.5-Sonnet | 29.1 ± 2.2            |
> | Qwen2-VL-72B-it   | 12.2 ± 1.6            |
> | Qwen2-VL-7B-it    | 4.4 ± 0.8             |
>
>
> We are also excited to highlight that AgentQuest will be open-sourced alongside a public leaderboard, enabling researchers to submit their models. You can see a fully anonymous mock version of the website here https://agentquestanon.github.io/. We hope the community will leverage this platform to address the limitations of current approaches and drive further innovation.

---

> > ### Author Response · Authors · 2024-11-18
> >
> > ### **[Q2] On the innovativeness of AgentQuest**
> > We respectfully disagree with the statement that AgentQuest does not qualify as innovative work. While aggregating existing environments, as the reviewer already identifies, it provides other major contributions. AgentQuest is a standardized framework that unifies a diverse set of Reinforcement Learning environments into a single, comprehensive benchmark designed to evaluate the agentic capabilities of LLMs and VLMs on long-horizon, interactive tasks, and to highlight specific capability gaps. It is worth noting that there is a long tradition of such comprehensive benchmarks in the field, such as BigBench, GLUE [2], SuperGLUE [3], HELM [4], various MLPerf benchmarks [5], and many more, which aggregate tasks to provide unified evaluation frameworks.
> >
> > Specifically, AgentQuest offers the following contributions:
> > 1. **An answer to an outstanding question in the literature:** “How do LLMs/VLMs perform on long-horizon Reinforcement Learning problems?”. We provide a potential answer to the question by evaluating the performance measures of a comprehensive set of models, both in text-only and text-image in Section 4.
> > 2. **A shared, unified prompting interface** for LLMs and VLMs to these RL environments. These environments lack an agreed-upon prompt to elicit responses from these models, which can be a great source of variance [6, 7]. AgentQuest introduces standard prompting conventions and mechanisms to handle invalid actions across them as discussed in Section 3.1 and 3.2, and shown in Appendix A.3, B.2, C.5, D.3, E.1, and F.4. This is crucial to exclude the impact of prompt engineering the environment observations on existing and future methods [8].
> > 3. **A gathering point of easy access.** AgentQuest offers a “plug-and-play” interface where researchers can submit their models once and receive a performance evaluation across all included environments. We provide instructions for submission in Section 2.2 and in the online documentation at https://agentquestanon.github.io/submit.html. This unified approach eliminates the need for custom integration with each environment, significantly lowering the barrier for researchers to evaluate and compare their methods, and saving time.
> > 4. **Additional evaluation metrics.** We provide fine-grained metrics for each environment in Section 3, including a novel data-informed progression metric for NetHack. Since some environments provide very little signal to evaluate these models, these metrics offer more meaningful insights into agent performance.
> >
> > We invite the reviewer to explore the updated version of our codebase, which exemplifies these integrations.
> >
> > ### **Conclusion**
> > We deeply value your feedback which has allowed us to refine AgentQuest further. We hope our clarifications demonstrate the significant effort and innovation behind AgentQuest, and address your concerns fully. Given these improvements we hope that you will increase the score for our paper, or point out what still stands in the way for a higher rating. Thank you for your time and effort in reviewing our work!
> >
> > ### **References**
> >
> > [1] https://www.anthropic.com/news/developing-computer-use
> >
> > [2]  Wang, Alex. "Glue: A multi-task benchmark and analysis platform for natural language understanding." arXiv preprint arXiv:1804.07461 (2018).
> >
> > [3] Wang, Alex, et al. "Superglue: A stickier benchmark for general-purpose language understanding systems." Advances in neural information processing systems 32 (2019)
> >
> > [4] Liang, Percy, et al. "Holistic evaluation of language models." arXiv preprint arXiv:2211.09110 (2022).
> >
> > [5] https://mlcommons.org/benchmarks/
> >
> > [6] Errica, Federico, et al. "What Did I Do Wrong? Quantifying LLMs' Sensitivity and Consistency to Prompt Engineering." arXiv preprint arXiv:2406.12334 (2024).
> >
> > [7] Sakai, Yusuke, et al. "Toward the Evaluation of Large Language Models Considering Score Variance across Instruction Templates." arXiv preprint arXiv:2408.12263 (2024).
> >
> > [8] Cao, Bowen, et al. "On the Worst Prompt Performance of Large Language Models." arXiv preprint arXiv:2406.10248 (2024).

---

> > > ### Author Response · Authors · 2024-11-22
> > >
> > > Dear Reviewer **HZcx**
> > >
> > > We hope this message finds you well.
> > >
> > > We have carefully addressed all your questions and concerns, and conducted additional evaluations of more agentic models as requested.
> > >
> > > As the rebuttal deadline is approaching soon, we kindly ask if you could strengthen support to the paper, or let us know if you have any further questions and we will reply swiftly.
> > >
> > > Kind regards,
> > >
> > > The Authors

---

> > > > ### Author Response · Authors · 2024-12-02
> > > >
> > > > Dear Reviewer **HZcx**
> > > >
> > > > We wanted to kindly remind you that today (December 2nd, EoD AoE) is the last day for reviewers to provide feedback. We can respond to any further questions or concerns until tomorrow.
> > > >
> > > > We have addressed your comments in detail, added new evaluations, and made significant updates. If there’s anything else you’d like us to clarify, please let us know.
> > > >
> > > > Kind regards,
> > > >
> > > > The Authors

---

### Official Review · Reviewer_pjfN · 2024-11-03

**Soundness:** 3
**Presentation:** 3
**Contribution:** 3
**Rating:** 6
**Confidence:** 4

**Summary:**

The paper introduces AgentQuest, a suite of six reinforcement learning environments for testing the agentic capabilities of long-context LLMs and VLMs. The benchmark includes diverse environments, ranging from simple tasks such as BabyAI, to complex tasks such as NetHack. The authors provide baseline evaluations of state-of-the-art LLMs/VLMs on AgentQuest using zero-shot prompting. The authors perform a qualitative analysis of the results across capabilities and identify an intriguing knowing-doing gap where the models cannot employ the knowledge they possess. Finally, the authors develop an open-source toolkit for AgentQuest.

**Strengths:**

- Comprehensive experiments. This paper is well-supported by comprehensive experiments and a diverve selection of environments that cover a wide range of agentic challenges.
- The proposed benchmark incorporates several typical environments and could facilitate rapid verification of LLM/VLM-based decision-making. Considering that existing LLM/VLM-based decision-making methods are evaluated by their own tasks or environments, such a unified benchmark will be beneficial and valuable to the whole community.
- The authors provide analysis into the experimental results on AgentQuest, including Spatial Reasoning, Long-term planning, Discovering and Leveraging Environment Dynamics, and Knowing-Doing Gap. The analysis provide insight into this problem, and could facilitate relevant research in this area.
- The writing is clear and easy to follow. The motivation is clearly presented.
- The code is provided to make the benchmark reproducible.

**Weaknesses:**

- More related decision-making environments should be discussed and compared. For instance, the authors include the Crafter in the benchmark, however there are also Minecraft environments such as Minedojo[1] and MineRL[2]. There are also LLM-based multi-agent cooperation benchmarks such as Overcook[3].
- More visualizations of the environment, agent behavior, and agent trajectories can be provided so that the analysis can be more intuitive.
- Does the AgentQuest support multi-agent LLM decision-making?
- How does AgentQuest deal with LLM hallucination? For instance, the LLM/VLM returns an invalid action.
- More cases should be provided for the limitations of VLMs.


[1] Fan, L., Wang, G., Jiang, Y., Mandlekar, A., Yang, Y., Zhu, H., ... & Anandkumar, A. (2022). Minedojo: Building open-ended embodied agents with internet-scale knowledge. Advances in Neural Information Processing Systems, 35, 18343-18362.

[2] Guss, W. H., Houghton, B., Topin, N., Wang, P., Codel, C., Veloso, M., & Salakhutdinov, R. (2019). Minerl: A large-scale dataset of minecraft demonstrations. arXiv preprint arXiv:1907.13440.

[3] Liu, J., Yu, C., Gao, J., Xie, Y., Liao, Q., Wu, Y., & Wang, Y. (2023). Llm-powered hierarchical language agent for real-time human-ai coordination. arXiv preprint arXiv:2312.15224.

**Questions:**

- Does the AgentQuest support multi-agent LLM decision-making?
- How does AgentQuest deal with LLM hallucination? For instance, the LLM/VLM returns an invalid action.

---

> ### Author Response · Authors · 2024-11-18
>
> We thank the reviewer for their thoughtful feedback and for acknowledging the strengths of our work, including the comprehensive experiments, diverse environments, insightful analyses, and reproducibility of our benchmark. We deeply appreciate your constructive suggestions, which have helped us refine and improve our paper.
>
> Below, we address each of your comments:
>
> ### **[Q1] More related environments should be compared**
> We appreciate the suggestion to discuss additional environments. A key design criterion for AgentQuest is that all included environments are solvable entirely by LLMs alone, with minimal reliance on computationally expensive visual processing. While environments such as MineDojo and MineRL are excellent for exploring open-ended agentic tasks, their high computational requirements and complex multimodal inputs would hinder the parallelizability and accessibility of our benchmark. In contrast, NetHack offers a similarly challenging environment but remains resource-friendly, making it ideal for AgentQuest.
>
> Regarding multi-agent collaboration environments like Overcooked, we agree these are valuable for evaluating teamwork. While AgentQuest currently focuses on single-agent capabilities to probe foundational decision-making skills, we don’t exclude that in the future we may support multi-agent environments as well. We have added a discussion on these environments and their tradeoffs to Section 5 of the revised paper.
>
> ### **[Q2] Visualization of the environment**
> We agree with the reviewer that additional visualizations would enhance the intuitiveness of our analyses. Unfortunately the constraints of a pdf paper restrict us from having videos or a large number of images.
>
>
> AgentQuest logs agents’ trajectories, making post-evaluation visualization straightforward. Upon open-sourcing, we will launch a website with a leaderboard linking to trajectory logs for all submissions. This will allow users to explore agent behaviors in detail. You can find a work-in-progress, anonymous version of this website here https://agentquestanon.github.io/, including some example gifs of the environment visuals.
>
> We believe this solution will make AgentQuest’s visual analysis accessible and comprehensive.
>
> ### **[Q3] Multi-Agent LLM decision Making**
> Thank you for raising this question! We interpret it in two ways and address both:
>
> Including multi-agent environments: AgentQuest currently focuses on single-agent scenarios to evaluate core decision-making skills such as long-term planning and systematic exploration without added complexity. While multi-agent environments are valuable, expanding to include them is a compelling future direction.
> Multi-Agent decision-making in single-agent environments: AgentQuest already supports multi-agent decision-making strategies applied to single-agent tasks. Researchers can integrate methods such as debate [6][7] or collaborative prompting. We anticipate that such techniques could significantly enhance performance, as noted in Section 6.
>
>
> We have added these considerations in the revised paper.
>
> ### **[Q4] Hallucinations and invalid actions in AgentQuest**
> AgentQuest handles hallucinated or invalid actions by:
> Providing feedback to the model that the action was invalid.
> Defaulting to a standard “do-nothing” action if supported by the environment, or a movement to “north/up” direction otherwise.
> Logging invalid actions in trajectory statistics for user analysis.
> These details have been added to the revised paper. Thank you for highlighting this.
>
>
> ### **[Q5] On the limitations of VLMs**
> We appreciate the reviewer’s suggestion to discuss further cases of the limitations of VLMs. From our experiments, we observe significant variability in VLM performance. While some models, like Llama 3.2, perform worse in VLM mode, often failing to integrate visual information into coherent decision-making, other models such as Gemini 1.5 Pro and the newly run Claude 3.5 sonnet, effectively utilize visual inputs.
>
> These findings align with challenges documented in recent work [1][2][3][4][5], which identify key limitations in VLMs, such as biases toward natural image-text pairs and optimization for image description rather than action-based reasoning. We have added the above and further considerations in Section 6 of the revised paper.
>
> ### **Conclusion**
> We thank the reviewer for their constructive feedback, which has allowed us to refine AgentQuest further. We hope that the revisions and additional analyses adequately address your concerns. We hope that based on this you will consider increasing the score for our paper, or point out what still stands in the way for a higher rating. Thank you for your time and effort in reviewing our work!

---

> > ### Author Response · Authors · 2024-11-18
> >
> > ### **References**
> >
> > [1] Tan, Weihao, et al. "Towards general computer control: A multimodal agent for red dead redemption ii as a case study." ICLR 2024 Workshop on Large Language Model (LLM) Agents. (2024).
> >
> > [2] Tong, Shengbang, et al. "Eyes wide shut? exploring the visual shortcomings of multimodal llms." Proceedings of the IEEE/CVF Conference on Computer Vision and Pattern Recognition. 2024.
> >
> > [3] Rahmanzadehgervi, Pooyan, et al. "Vision language models are blind." arXiv preprint arXiv:2407.06581 (2024).
> >
> > [4] Zang, Yuhang, et al. "Overcoming the Pitfalls of Vision-Language Model Finetuning for OOD Generalization." arXiv preprint arXiv:2401.15914 (2024).
> >
> > [5] Guan, Tianrui, et al. "HallusionBench: An Advanced Diagnostic Suite for Entangled Language Hallucination and Visual Illusion in Large Vision-Language Models." arXiv preprint arXiv:2310.14566 (2023).
> >
> > [6] Du, Yilun, et al. "Improving factuality and reasoning in language models through multiagent debate." arXiv preprint arXiv:2305.14325 (2023).
> >
> > [7] Khan, Akbir, et al. "Debating with more persuasive llms leads to more truthful answers." arXiv preprint arXiv:2402.06782 (2024).

---

> > > ### Author Response · Authors · 2024-11-22
> > >
> > > Dear Reviewer **pjFN**
> > >
> > > We hope this message finds you well.
> > >
> > > We have carefully addressed all your questions and comments, and have revised the paper to incorporate your suggestions. As the rebuttal deadline is approaching soon, we kindly ask if you could strengthen support to the paper, or let us know if you have any further questions and we will reply swiftly.
> > >
> > > Kind regards,
> > >
> > > The Authors

---

> > > > ### Comment · Reviewer_pjfN · 2024-11-27
> > > > **Official Comment by Reviewer pjfN**
> > > >
> > > > Thank you for your response. It would be a more complete work if the authors expanded their work to multi-agent scenarios. Therefore I choose to keep my rating.

---

> ### Author Response · Authors · 2024-11-28
>
> Dear Reviewer, we would like to offer a contrasting view on why the absence of multi-agent environments does not limit the completeness of our benchmark. While we agree that multi-agent research is extremely important, it is orthogonal and complementary to the goals of this benchmark. Multi-agent environments are valuable for studying coordination and communication, whereas AgentQuest is designed to probe foundational agentic capabilities such as sequential decision-making, long-term planning, spatial reasoning, and systematic exploration.
>
> We are not claiming that AgentQuest addresses all potential agentic challenges. Instead, it provides a rigorous and focused framework for evaluating core capabilities of LLMs and VLMs. Adding further multi-agent environments would increase complexity and resource requirements for running the benchmark, potentially hindering accessibility for the broader research community, while targeting different research questions.
>
> That being said, we value your concerns and have conducted additional experiments to explore VLM limitations. Specifically, we developed a 3D Minecraft-style rendering of the Crafter environment to investigate whether enhanced visual representations improve performance. We invite you to review these findings, which are included in our updated shared rebuttal and detailed in Appendix B.3 of the revised paper.
>
> We hope this discussion has provided clarity and convinced you to reconsider your view of AgentQuest as a complete benchmark in its current scope. Please let us know if you have any further thoughts or feedback, and we will get back to you swiftly.

---

### Official Review · Reviewer_LP97 · 2024-11-04

**Soundness:** 3
**Presentation:** 3
**Contribution:** 2
**Rating:** 8
**Confidence:** 3

**Summary:**

The paper presents AgentQuest, a benchmark to evaluate LLMs and VLMs on complex, long-horizon tasks. By consolidating multiple reinforcement learning environments (e.g., NetHack, TextWorld, BabyAI), AgentQuest provides a unified testbed for assessing skills like spatial reasoning, long-term planning, and exploration. Findings show that while LLMs perform adequately on simpler tasks, they struggle with complex ones, especially when handling visual inputs. AgentQuest highlights gaps between current model capabilities and human-level performance, encouraging further research to enhance agentic qualities in AI.

**Strengths:**

1. AgentQuest integrates diverse environments, testing a wide range of agentic skills from navigation to complex reasoning, making it a well-rounded benchmark.

2. The benchmark’s emphasis on agentic capabilities addresses key limitations in current LLMs and VLMs, which are crucial for autonomous AI applications.

3. The benchmark’s metrics allow for detailed analysis of model performance, identifying specific areas like spatial reasoning and long-term planning where models struggle.

**Weaknesses:**

1. It is weird that VLMs underperform when visual information is included. It indicates that current architectures may not be well-suited for complex vision-based reasoning. Or the current prompt is not suitable for the model. I think this point deserves a careful study.

2. More long-context models should be included for evaluation, for example Qwen2, Llama-3-8B-Instruct-80K, LongAlpaca-7B, LongChat-v1.5-7B-32k.

**Questions:**

1. How might the benchmark adapt to new multimodal architectures that integrate more advanced vision processing, such as video-based reasoning?

2. Could the “knowing-doing” gap (where models have knowledge but fail in practical application) be addressed by incorporating memory or reinforcement learning methods in future iterations of the benchmark?

---

> ### Author Response · Authors · 2024-11-18
>
> We thank the reviewer for the positive comments on our work and for recognizing that AgentQuest is a well-rounded benchmark that addresses key limitations in current LLM/VLM agentic capabilities with metrics allowing for a detailed analysis on model performance.
>
> We address your comments below:
>
> ### **[Q1] On the underperformance of VLMs**
> We agree with the reviewer that the underperformance of VLMs when visual information is included deserves careful investigation. This challenge, also noted in recent work [1][2][3][4][5], is an important area for ongoing research and we have conducted more experiments to explore it further.
>
> Specifically, in our new experiments with Claude 3.5 sonnet on AgentQuest, we observed improved performance in the VLM mode compared to the language-only mode using identical prompts. This suggests that some VLMs can utilize visual information effectively. Claude-3.5-Sonnet's superior performance could be attributed to its training on computer interaction [6], which likely enhances its capabilities in agentic tasks like AgentQuest with visual inputs. We note that Gemini 1.5 Pro also was able to properly utilize visual inputs, with its performance in vision-language mode being comparable to language-only mode. We believe AgentQuest will be a valuable benchmark to measure progress in VLMs going forward.
>
> ### **[Q2] Including more long-context models**
> We thank the reviewer for suggesting additional models to evaluate. In response, we expanded our evaluations to include several additional models:
> - Qwen2.5 (7B/72B)-instruct,
> - Qwen2 VL (7B/72B)-instruct,
> - Claude-3.5-sonnet,
> - Llama-3.1/3.2 variants (1B/3B/8B)-it.
> You can find the results in our updated paper or on our fully anonymous submission website: https://agentquestanon.github.io/ and in the table below:
>
> LLM mode
> | Model               | Progress ± SE         |
> |---------------------|-----------------------|
> | Claude-3.5-Sonnet   | 30.0 ± 2.0            |
> | Qwen2.5-72B-it      | 16.2 ± 1.6            |
> | Llama-3.1-8B-it     | 14.1 ± 1.5            |
> | Qwen2-VL-72B-it     | 12.8 ± 1.6            |
> | Llama-3.2-3B-it     | 8.5 ± 1.1             |
> | Qwen-2.5-7B-it      | 7.8 ± 1.1             |
> | Llama-3.2-1B-it     | 6.3 ± 1.0             |
> | Qwen2-VL-7B-it      | 3.7 ± 0.8             |
>
> VLM mode
> | Model             | Progress ± SE         |
> |-------------------|-----------------------|
> | Claude-3.5-Sonnet | 29.1 ± 2.2            |
> | Qwen2-VL-72B-it   | 12.2 ± 1.6            |
> | Qwen2-VL-7B-it    | 4.4 ± 0.8             |
>
>
> We also evaluated LongAlpaca-7B and LongChat-v1.5-7B-32k as suggested. However, these models failed to produce meaningful evaluations due to their design as chat-oriented models rather than true instruction-following models. Specifically, these models are fine-tuned from Llama 2 chat variants and exhibited behavior better suited for conversational tasks, often generating verbose and irrelevant responses instead of reliably outputting the required actions. As a result, their trajectories consistently scored 0% across all environments in the benchmark.
>
> Our additional findings highlight the crucial role of robust instruction-following capabilities for successful performance in AgentQuest. Models must be capable of reading the rules of the game, understanding the action space, and outputting actions to navigate and complete the tasks evaluated by the benchmark. We have added this consideration to Section 3.1 of the revised paper.
>
>
> ### **[Q3] Adapting the benchmark to new multimodal architectures**
> AgentQuest is designed to be extensible and forward-compatible to new architectures, as shown in section 2.2. The AgentQuest team is going to support and keep up with newly released multi-modal models, making it as easy as possible for future users. We are excited for users to submit models utilizing such type of video reasoning, however, we also recognize that currently it is very expensive to use videos with multimodal LLMs. We do plan on supporting video observations once prominent models with efficient video-processing capabilities become available, as highlighted in section 6 of the revised paper.

---

> > ### Author Response · Authors · 2024-11-18
> >
> > ### **[Q4] Addressing the “knowing-doing” gap**
> > We appreciate the reviewer’s suggestion to address the knowing-doing through memory mechanisms and reinforcement learning techniques. We have incorporated a discussion on this consideration to section 6 of the paper. We also want to emphasize that the models/agents submitted by the AgentQuest team are supposed to be baselines on which future research can be built upon. There is no gold standard method on how to perform this yet, and progress in this active area of research is precisely what we want to encourage with this benchmark. Specifically, AgentQuest is designed to enable:
> > Advanced Reasoning Strategies: techniques like chain-of-thought prompting can encourage deeper reasoning.
> > External Memory Modules: To help models utilize knowledge more effectively over long horizons
> > In-Context Learning and Few-Shot Prompting: AgentQuest’s API supports this, which will be a valuable tool for research going forward.
> >
> >
> > ### **Conclusion**
> > We appreciate the reviewer’s thoughtful feedback. We believe that the improvements we’ve made significantly strengthened our paper. We hope that based on this you will consider increasing the score for our paper, or point out what still stands in the way for a higher rating. Thank you once again for your constructive feedback!
> >
> > ### **References**
> >
> > [1] Tan, Weihao, et al. "Towards general computer control: A multimodal agent for red dead redemption ii as a case study." ICLR 2024 Workshop on Large Language Model (LLM) Agents. (2024).
> >
> > [2] Tong, Shengbang, et al. "Eyes wide shut? exploring the visual shortcomings of multimodal llms." Proceedings of the IEEE/CVF Conference on Computer Vision and Pattern Recognition. 2024.
> >
> > [3] Rahmanzadehgervi, Pooyan, et al. "Vision language models are blind." arXiv preprint arXiv:2407.06581 (2024).
> >
> > [4] Zang, Yuhang, et al. "Overcoming the Pitfalls of Vision-Language Model Finetuning for OOD Generalization." arXiv preprint arXiv:2401.15914 (2024).
> >
> > [5] Guan, Tianrui, et al. "HallusionBench: An Advanced Diagnostic Suite for Entangled Language Hallucination and Visual Illusion in Large Vision-Language Models." arXiv preprint arXiv:2310.14566 (2023).
> >
> > [6] https://www.anthropic.com/news/developing-computer-use

---

> > > ### Author Response · Authors · 2024-11-22
> > >
> > > Dear Reviewer **LP97**
> > >
> > > We hope this message finds you well.
> > >
> > > We have carefully addressed all your questions and comments and conducted additional evaluations of models as requested.
> > >
> > > As the rebuttal deadline is approaching soon, we kindly ask if you could strengthen support to the paper, or let us know if you have any further questions and we will reply swiftly.
> > >
> > > Kind regards,
> > >
> > > The Authors

---

> > > > ### Comment · Reviewer_LP97 · 2024-11-25
> > > > **Reply to Authors**
> > > >
> > > > Thanks for your detailed response. My concerns are almost addressed. This rebuttal is insightful and I will keep the original rate.

---

> > > > > ### Author Response · Authors · 2024-11-25
> > > > >
> > > > > Thank you for your feedback. Since you mentioned that your concerns are "almost addressed," could you kindly clarify any remaining issues? We would be happy to address them promptly.

---

> ### Author Response · Authors · 2024-11-28
>
> Dear Reviewer, to further address your concerns on VLM performance, we conducted additional experiments using a 3D Minecraft-like rendering of the Crafter environment developed by us. The findings are included in our updated shared rebuttal and detailed in Appendix B.3 of the revised paper. We kindly invite you to review these updates and share any further thoughts.

---

> > ### Comment · Reviewer_LP97 · 2024-11-28
> > **Reply to Authors**
> >
> > Thanks for your efforts on improving the quality of this paper. I think this is a solid paper. I will improve the rating.

---

> > > ### Author Response · Authors · 2024-11-28
> > >
> > > Dear Reviewer, thank you for recognizing our efforts to improve the quality of the paper. Your constructive feedback has been invaluable, and we greatly appreciate your support.

---

### Author Response · Authors · 2024-11-18

We thank the reviewers for their thoughtful feedback, which has been invaluable in enhancing the quality of our work. We are glad that reviewers recognized that AgentQuest is a timely (Reviewer **SuiJ**) well-rounded (Reviewer **LP97**), and comprehensive benchmark (Reviewers **pjFN**, **HZcx**), encompassing a diverse selection of environments testing a wide range of skills (Reviewers **LP97**, **pjFN**), with metrics allowing for detailed analysis of model performance (Reviewers **LP97**, **HZcx**). Reviewers also agreed that AgentQuest could facilitate rapid verification and spark more research (Reviewers **pjFN**, **SuiJ**) that would be beneficial to the whole research community (Reviewer **pjFN**).

To further strengthen the comprehensiveness of our evaluation, we have now added the following models, as also requested by Reviewers **LP97** and **HZcx**
- Claude 3.5 Sonnet,
- Llama-3.1/3.2 variants (1B/3B/8B)-it.
- Qwen2.5 (7B/72B)-instruct,
- Qwen2 VL (7B/72B)-instruct,

You can view all the results in our updated paper, as well as in our fully anonymous leaderboard https://agentquestanon.github.io/ which will be released to the public upon open-sourcing.

One common question raised by reviewers relates to the underperformance of some Vision-Language Models (VLMs) in AgentQuest. Reviewer **LP97** noted that this deserves careful study, Reviewer **pjFN** suggested providing additional cases, and Reviewer **SuiJ** highlighted the potential impact of out-of-distribution (OOD) visual inputs. We fully agree that understanding VLM performance is crucial and appreciate the opportunity to clarify.

Not all VLMs underperform in AgentQuest. While some models, such as Llama 3.2, struggle to effectively leverage visual inputs in decision-making tasks, other models, such as Gemini 1.5 Pro and the newly evaluated Claude 3.5 Sonnet, demonstrate improved performance in VLM mode, effectively utilizing added visual inputs. This diversity of performance highlights the importance of AgentQuest in identifying specific strengths and weaknesses in current architectures and guiding future research.

AgentQuest, as a benchmark, is designed to discover precisely these kinds of gaps in model capabilities. Its role is not only to evaluate current models but also to identify areas where improvement is needed. These findings align with recent work [1][2][3][4][5], which has documented similar challenges with VLMs. Thus, we view this as a strength of AgentQuest, highlighting its value as a tool for advancing research in this domain.

We have more thoroughly addressed each reviewer’s individual concerns in their respective rebuttals.

Once again, we thank the reviewers again for their constructive feedback, which has enabled us to strengthen our paper. We hope that the enhancements and clarifications we have made have addressed your questions and that the reviewers will consider increasing their scores to further support the acceptance of our paper.

Kind regards,

The Authors


### **References**
[1] Tan, Weihao, et al. "Towards general computer control: A multimodal agent for red dead redemption ii as a case study." ICLR 2024 Workshop on Large Language Model (LLM) Agents. (2024).

[2] Tong, Shengbang, et al. "Eyes wide shut? exploring the visual shortcomings of multimodal llms." Proceedings of the IEEE/CVF Conference on Computer Vision and Pattern Recognition. 2024.

[3] Rahmanzadehgervi, Pooyan, et al. "Vision language models are blind." arXiv preprint arXiv:2407.06581 (2024).

[4] Zang, Yuhang, et al. "Overcoming the Pitfalls of Vision-Language Model Finetuning for OOD Generalization." arXiv preprint arXiv:2401.15914 (2024).

[5] Guan, Tianrui, et al. "HallusionBench: An Advanced Diagnostic Suite for Entangled Language Hallucination and Visual Illusion in Large Vision-Language Models." arXiv preprint arXiv:2310.14566 (2023).

---

> ### Author Response · Authors · 2024-11-28
>
> Dear Reviewers,
>
> We conducted a new set of experiments to answer common questions raised by the reviewers related to the underperformance of some VLMs on AgentQuest. Specifically, we investigated whether the visual representation could affect model performance in the Crafter environment.
>
> To address this question systematically, **we developed an augmented version of the Crafter environment that renders scenes in 3D using Minecraft-style models and textures**, instead of the original 2D tiled textures, see Figure 4 in the revised version of the paper for visualization. We hypothesized that VLMs might perform better with 3D renderings, given that natural images which were probably a major part of the training dataset, represent three-dimensional environments.
>
> Interestingly, our results revealed that the switch to 3D rendering did not lead to improved performance. We observed that in several instances, the models performed slightly worse in the 3D-rendered environment compared to the original 2D version. This sheds new light on the existing results. These results suggest that the underperformance of VLMs in our benchmark may stem from more fundamental limitations in their architecture or reasoning capabilities, rather than from the specific visual representation format. The models struggle with complex vision-based reasoning tasks regardless of whether the visual input matches the distributions they were likely trained on.
>
> These new findings have been included in Appendix B.3 of the revised paper, and invite you to explore the 3D Crafter renderings and results presented there. While this sheds new light on existing results, we acknowledge that further investigation into this area is warranted.
>
> We hope these updates address your concerns and look forward to hearing your thoughts.

---

### Meta-Review · Area_Chair_8DJt · 2024-12-21

**Metareview:**

The paper proposes a benchmark to evaluate LLMs and VLMs on complex, long-horizon tasks and evaluates it in multiple reinforcement learning environments. While the finding the somewhat expected that LLMs perform well on simple tasks and struggle on tasks with visual inputs, the paper is a good step toward benchmarking the capabilities of the existing models. All of the reviewers recognize the contribution of this work. The main weaknesses are requesting evaluation on more models and request for more analysis on the result. The authors will also make the benchmark publicly available with a leaderboard. Overall, the AC thinks there is enough contribution from the paper, however, the paper's finding is somewhat expected and there isn't a strong technical contribution component.

**Additional Comments On Reviewer Discussion:**

All but one reviewer participated in the discussion period. The author addressed the reviewers' concerns by providing additional evaluations using more models and clarifying some of the requested details. Overall, the reviewers were satisfied with the response, and the AC agrees that this work is likely to have a positive impact on the community, although the findings are somewhat expected.

---

### Decision · Program_Chairs · 2025-01-22

Accept (Poster)